# Inflammatory bone marrow signaling in pediatric acute myeloid leukemia distinguishes patients with poor outcomes

Hamid Bolouri [1,5,6] ✉, Rhonda E. Ries [2,5], Alice E. Wiedeman [3,5], Tiffany Hylkema[2], Sheila Scheiding [3], Vivian H. Gersuk [1], Kimberly O'Brien[1], Quynh-Anh Nguyen[1], Jenny L. Smith [2,4], S. Alice Long [3] & Soheil Meshinchi[2,6] ✉

High levels of the inflammatory cytokine IL-6 in the bone marrow are associated with poor outcomes in pediatric acute myeloid leukemia (pAML), but its etiology remains unknown. Using RNA-seq data from pre-treatment bone marrows of 1489 children with pAML, we show that > 20% of patients have concurrent IL-6, IL-1, IFNα/β, and TNFα signaling activity and poorer outcomes. Targeted sequencing of pre-treatment bone marrow samples from affected patients ($n = 181$) revealed 5 highly recurrent patterns of somatic mutation. Using differential expression analyses of the most common genomic subtypes (~60% of total), we identify high expression of multiple potential drivers of inflammation-related treatment resistance. Regardless of genomic subtype, we show that JAK1/2 inhibition reduces receptor-mediated inflammatory signaling by leukemic cells in-vitro. The large number of high-risk pAML genomic subtypes presents an obstacle to the development of mutation-specific therapies. Our findings suggest that therapies targeting inflammatory signaling may be effective across multiple genomic subtypes of pAML.

In spite of its heavy toll on families and young lives, treatments for pediatric Acute Myeloid Leukemia (pAML) lag far behind other leukemias and the 10-year survival rate for children with pAML is <50%[1]. This is due, in part, to the fact that pAML is a highly heterogeneous disease comprising many clinical, genomic, epigenomic, and transcriptional subtypes[2] (Supplementary Fig. 1). While genomic/precision medicine has delivered many successes over the past decade[3,4], development of targeted therapies for each of these rare, poor-prognosis pAML subtypes may be economically prohibitive. To address this challenge, we set out to identify dysregulated pathways that are shared across multiple pAML subtypes and mark poorer

clinical outcomes. By spanning multiple disease subtypes, treatments targeting such pathways can overcome the cost-per patient barrier and deliver treatments to those most in need.

Inflammatory signaling is a hallmark of cancer[5,6]. Accordingly, the pro-inflammatory IL-6 signaling pathway is a well-established contributor to many cancers[7]. IL-6 synergizes with IL-3 to drive the proliferation of hematopoietic stem cells[8], and increased bone marrow (BM) IL-6 levels have been shown to correlate with treatment resistance and poor outcomes in pediatric[9] and adult AML[10,11] as well as in other blood malignancies[12,13]. IL-6 signals through JAK1, JAK2, and STAT3[14], and JAK2/STAT3 signaling is increased in the blood

[1]Center for Systems Immunology, Benaroya Research Institute, 1201 9th Ave, Seattle, WA, USA. [2]Clinical Research Division, Fred Hutchinson Cancer Research Center, 1100 Fairview Ave N, Seattle, WA, USA. [3]Center for Translational Immunology, Benaroya Research Institute, 1201 9th Ave, Seattle, WA, USA. [4]Present address: Research Scientific Computing, Seattle Children's Research Institute, 818 Stewart Street, Seattle, WA, USA. [5]These authors contributed equally: Hamid Bolouri, Rhonda E. Ries, Alice E. Wiedeman. [6]These authors jointly supervised this work: Hamid Bolouri, Soheil Meshinchi. ✉e-mail: HBolouri@BenaroyaResearch.org; SMeshinc@FredHutch.org

progenitor/stem cells of high-risk adult AML[15]. Consistent with a role for IL-6 signaling in treatment resistance, inhibition of JAK/STAT signaling sensitizes AML blasts to treatment with other drugs[16], as well as reducing the proliferation rate of adult AML cells[17]. Moreover, the IL-6 neutralizing antibody Siltuximab increased the survival of mice engrafted with adult AML blasts[18]. Whether the IL-6 signaling pathway is activated in pAML leukemic cells, whether it is associated with specific subtypes of pAML, whether it has distinct roles in pediatric versus adult patients, and whether it is the key upstream driver of poor outcomes in pAML remain unknown.

To address these questions, we analyzed RNA-seq data from ~1500 pAML patients enrolled in 2 large-scale Children's Oncology Group (COG) studies, and verified our findings through additional genomic, epigenomic, flow cytometry, and cytometry by time of flight (CyTOF) analyses on selected pAML samples. Here, we show that a subset of pAML patients with higher levels of IL-6 signaling in their pre-treatment bone marrow samples, have poorer 2-year event-free and overall survival, and higher inflammatory cytokine signaling via the IFNα/β, IL-1, and TNFα pathways. As such, combination therapies that counter these signal transduction pathways may help a relatively large (> 20%), genomically-diverse proportion of pAML patients avoid treatment resistance and poor outcomes.

## Results

### Higher IL-6 signaling occurs in > 20% of pAML patients and is associated with worse outcomes

Comparing bulk RNA-seq data from 1,275 pAML BM aspirates collected at diagnosis prior to treatment (COG study AAML1031, NCT01371981) to 45 age-matched healthy normal bone marrow (NBM) controls (Supplementary Fig. 2), we found that mRNA levels of *IL-6* and its receptor subunit *IL-6R* are rarely higher in pAML than in NBM (Fig. 1a). However, *IL-6* and *IL-6R* mRNA levels were bimodally distributed across the pAML cohort, suggesting 2 potentially distinct patient populations (Fig. 1b). Using Gaussian mixture modeling, we identified subsets of samples with higher expression of either *IL-6* or *IL-6R* (red bars in Fig. 1b) and selected patients in the highest-expressing quartile in each higher-expression group as having unusually high levels of one or both genes (287 patients of whom 7 had high levels of both *IL-6* and *IL-6R*, 22.5% of the cohort). We refer to this group as the "high-IL6/R" group. For comparison, we also selected a reference group of patients with

below-median expression of both *IL-6* and *IL-6R* across all patients in the cohort ("low-IL6/R" group, 306 patients, 24% of the cohort).

IL-6 is associated with chemotherapy and multi-drug resistance in diverse cancers[19]. Consistent with these findings, high-IL6/R pAML patients had significantly worse event free survival (EFS) and overall survival (OS) within 2 years of diagnosis (proportion of high-IL6/R patients alive at 2 years = 0.591. 95% Confidence Interval = (0.558, 0.677). The proportion of low-IL6/R patients alive at 2 years = 0.716. 95% Confidence Interval = (0.708, 0.808), Fig. 1c, 5-year plots in Supplementary Fig. 3). High- and low-IL6/R samples had similar percentages of blasts (means 62.0, 67.9 respectively, Fig. 1d), male-to-female ratios (1.1, 1.1 respectively), and AAML1031 Study Arm participation (Arm A 32% vs. 40%, Arm B 31% vs. 38%, Arm C 42% vs. 52% respectively, Fisher's Exact Test P ~ 1), suggesting the observed differences are not confounded by these factors. However, high-IL6/R samples had significantly higher average expression levels of a proliferation-marker gene set (99 genes, Mann–Whitney *U* Test P < 1E−24, Supplementary Fig. 4).

To identify pathways and processes associated with higher levels of IL-6 and IL-6R in high- versus low-IL6/R samples, we carried out a series of gene set enrichment analyses contrasting these 2 patient groups (Fig. 1e and Supplementary Fig. 5). As expected, compared to low-IL6/R samples, high-IL6/R samples were highly enriched for an IL-6 signaling gene set (87 genes, Mann–Whitney U Test P < 1E−51). Unexpectedly, we found that, compared to low-IL6/R samples, high-IL6/R samples are also highly enriched for gene sets marking Type 1 interferon signaling (66 genes, Mann–Whitney *U* Test P < 1E−17), IL-1 signaling (149 genes, Mann–Whitney *U* Test P < 1E−37), and TNFα signaling (197 genes, Mann–Whitney *U* Test P < 3E−22).

Although IL-6, IFNα/β, IL-1, and TNFα signaling co-occur in approximately two thirds of all samples (Supplementary Fig. 6), we found that donor selection based on the expression levels of *IL-6* and *IL-6R* was a better predictor of EFS and OS than sample selection using IFNα/β or IL-1 signaling gene sets (Supplementary Figs. 7–10), suggesting IL-6 signaling is the most likely driver of the observed poorer outcomes.

To verify the generality of our findings in an independent cohort, we used our previously published bulk BM RNA-seq data for 214 COG AAML0531 study (NCT00372593) participants[2]. Applying the same selection criteria to this cohort as used for AAML1031, we identified 43 (20.1%) high-IL6/R and 59 (27.6%) low-IL6/R samples (Supplementary

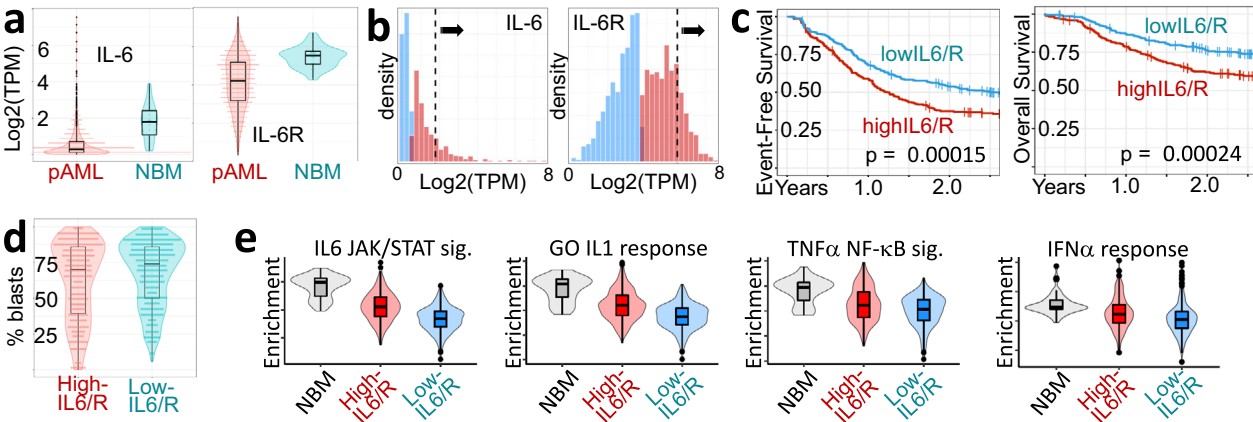

**Fig. 1 | Pediatric AML diagnostic samples with higher IL-6 or IL-6R expression have higher multi-cytokine signaling in the bone marrow (BM) and poorer 2-year outcomes than patients with low levels of bone-marrow IL-6 and IL-6R mRNA. a** pediatric AML BM *IL-6* and *IL-6R* expression levels are rarely higher than in normal BM. **b** Gaussian mixture deconvolution of *IL-6* and *IL-6R* expression into low and high groups (blue and red histograms, respectively). The top 25% of samples in the higher-expression groups (black arrows and dashed lines) were selected as having unusually high IL6/IL6R expression levels. **c** Kaplan–Meier Event-Free and

Overall Survival plots for high- vs. low-IL6/R AML. 2-year log-rank p-values are indicated in each plot. **d** High- and low-IL6/R samples have similar fractions of blast cells. **e** Relative Single Sample Gene Set Enrichment Analysis (ssGSEA) scores for the IL-6 signaling gene set from MSigDB (gsea-msigdb.org), the Gene Ontology 'IL-1 Response' gene set (GO:0071347), and the TNFα signaling and IFNα response gene sets from MSigDB. Boxplots show the median and the upper and lower central-quartiles. The expected range of the data is indicated by whiskers. In all panels n = 287 for high-IL6/R group, n = 306 for the low-IL6/R group, and n = 45 for NBM.

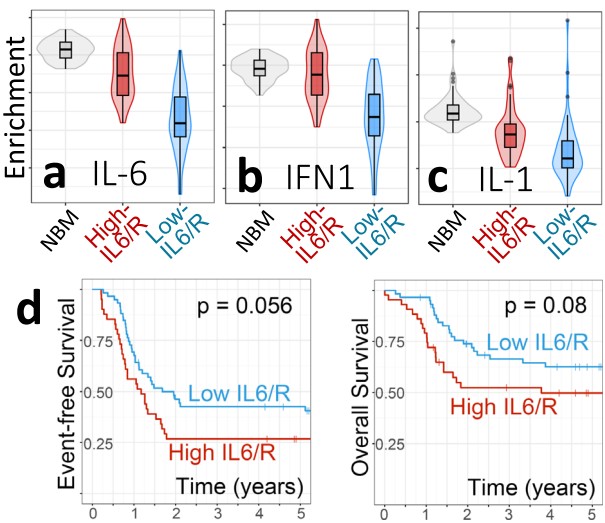

**Fig. 2 | Verification of findings in an independent cohort using the same gene sets and scoring as in the discovery cohort (Fig. 1, Supplementary Fig. 2B).**
**a** IL-6 signaling, **b** Type 1 IFN signaling, **c** IL-1 signaling. Shown are gene set mean expression levels in high- vs. low-IL6/R samples. **d** Comparison of 5-year Event-Free and Overall Survival for high- vs. low-IL6/R patients. Log-rank $p$-values are shown. Boxplots show the median and the upper and lower central-quartiles. The expected range of the data is indicated by whiskers. In all panels $n = 43$ for high-IL6/R group, $n = 59$ for the low-IL6/R group., and $n = 62$ for NBM.

Fig. 11). Using the same gene sets as for AAML1031, we again found higher activity scores for IL-6, IFNα/β and IL-1 signaling (Mann–Whitney $U$ Test $P$ values <1E−12, <1E−5, and <1E−7 respectively, Fig. 2a–c. Supplementary Fig. 12). Five-year EFS and OS both exhibited a trend toward worse outcomes in high-IL6/R patients compared to low-IL6/R patients (Fig. 2d). Consistent with a role of for IL-6 in treatment resistance, two-year EFS and OS were significantly worse for high-IL6/R patients compared to low-IL6/R patients log-rank $P = 0.028$ and 0.014 respectively, with the proportion of high-IL6/R alive at 2 years = 0.512, 95% Confidence Interval = (0.392, 0.700), the proportion of low-IL6/R alive at 2 years = 0.738, 95% Confidence Interval = (0.633, 0.861)). Together, these findings suggest that higher levels of multi-cytokine inflammatory signaling in high- versus low-IL6/R pAML mark poorer 2-year outcomes in > 20% of pediatric AML.

### High-IL6/R pAML samples have distinct immunological, transcriptomic, demographic, and genomic profiles

To better understand the etiology of multi-cytokine signaling in pAML, we sought to identify the key features of high-IL6/R pAML that set it apart from low-IL/R pAML. We started by asking whether the higher expression of inflammatory cytokine signaling in high- versus low-IL6/R pAML reflects broader immune function differences. Of 1793 immune genes downloaded from ImmPort.org 1506 are expressed in our RNA-seq data and collectively segregate high- and low-IL6/R pAML and NBM into distinct groups (Fig. 3a). Thus, there are genome-wide immune-state differences between high- vs. low-IL6/R pAML, and between these pAML subsets and NBM.

Using genome-wide differential expression analysis, we found transcriptional differences between high- and low-IL6/R pAML were not limited to the immune system. At a false discovery rate (FDR)-adjusted $P$ value <0.05 and absolute fold-change > 0.25, out of a total of 51,573 measured transcripts, 15,609 (~30%) were differentially expressed between high-IL-6/R and low-IL6/R pAML, and 14,423 transcripts (~28%) were differentially expressed between high-IL6/R and NBM samples (Supplementary Data 1 and 2). Thus, transcriptional differences between high- and low-IL6/R pAML, and between high-IL6/R pAML and NBM, are pervasive and genome-wide.

To infer the key cancer and immune-associated genes underlying these differences, we used iterative feature selection from a collection of 4015 immune, epigenetic, and AML-related genes. We identified a set of 82 cancer and immune-related genes (Supplementary Data 3) that divide high- and low-IL6/R pAML samples into high-, medium-, and low-IL6/R clusters by unsupervised hierarchical clustering (clusters 1–3 respectively, Fig. 3B, Supplementary Fig. 13). More than 98% of Cluster1 samples are in the high-IL6/R group, while Cluster3 samples are highly concordant with our reference low-IL6/R samples (Supplementary Figs. 14–16). These transcriptionally-derived clusters revealed two surprising clinical characteristics of high-IL6/R pAML. Firstly, 48% of the donors in Cluster1 of Fig. 3b had clinically-identified chromosomal translocations impacting the *MLL (KMT2A)* gene, compared to only 7% in the low-IL6/R Cluster3. Consistent with this finding, high-IL6/R pAML BM samples are > 2-fold enriched for the French-American-British (FAB) M4 and M5 and WHO "AML, not otherwise categorized: Acute monoblastic/acute monocytic leukemia" and "AML, not otherwise categorized: Acute monoblastic/acute monocytic leukemia" classification categories (Fisher's Exact Test $P = 3.9E−20$, odds ratio 3.76, 95% confidence interval: 2.8, 5.0, comparing high-IL6/R samples ($n = 287$) to all other AAML1031 donors ($n = 1063$)). In contrast, 43.8% of the low-IL6/R Cluster3 samples had no clinically-detected chromosomal abnormalities

High-IL6/R pAML patients were significantly younger than low-IL6/R pAML (median age at diagnosis = 8.4 and 11.7 years, respectively, Mann–Whitney $U$ Test $P$ value = 1.6E−05). 58.6% of the patients in Cluster1 are <10 years old and 41% are infants aged ≤ 3 years old. In contrast, only 16% of the low-IL6/R Cluster3 patients are infants (2-sided test of proportions $P = 1.9E−08$, Supplementary Fig. 17).

To establish whether high-IL6/R pAML samples have additional shared genomic aberrations, we analyzed targeted sequencing data (Supplementary Data 4, summarized in Fig. 3c) for the 181 patients in the high-IL6/R transcriptional cluster described above (Cluster1, Fig. 3b). Targeted sequencing data for a panel of 580 recurrent single nucleotide alterations and short insertion/deletions (indels), and 15 translocations previously identified by the NCI TARGET AML Project[2], revealed that 142 of 181 Cluster1 high-IL6/R samples (78.5%) had recurrent translocations. 91 samples (50.3%) had *MLL (KMT2A)* translocations, 55 (30.4%) had RAS point mutations, and 36 (19.9%) had *FLT3* point mutations (Fisher's Exact Test $P$ values 9.6E−29, 0.004, and 0.004, respectively when compared to Cluster3). Furthermore, 77.8% of the *FLT3* point mutations were demonstrated to be activating mutations by functional assays (Supplementary Data 4). Only 1 *FLT3* internal tandem duplication (*FLT3*-ITD) co-occurred with an MLL-translocation. In contrast, among the Cluster1 samples without *MLL*-rearrangements, 21 (23.3%) had *FLT3*-ITD.

Consistent with the widespread occurrence of *MLL* gene-fusions in high-IL6/R samples, the mean expression of the MSigDB (gsea-msigdb.org/gsea/msigdb) "MLL-MLLT3 Fusion Target Genes" gene set was significantly higher in high-IL6/R pAML compared to low-IL6/R pAML (Mann–Whitney U Test $P < 1E−44$).

The patterns of co-occurrence and mutual-exclusion of the above recurrent somatic mutations divide the patients into distinct genomic groups. More than 93% of high-IL6/R pAML samples fall into one of 5 genomic subtypes (indicated by Group 1-5 color ribbons at the top of Fig. 3c). Patients in Groups 1 and 2 have chromosomal rearrangements resulting in the fusion of the *MLL/KMT2A* gene to one of 8 well-known *MLL* fusion partners. Group 1 patients additionally have *RAS* or *FLT3* point mutations, both of which impact proliferation. The third high-IL6/R genomic group (indicated by the broken green bar at the top of Fig. 3c) is characterized by the co-occurrence *RAS/FLT3* mutations with various pAML-associated chromosomal translocations, including chromosome 16 inversions, and *RUNX1* and *NUP98* fusions. Group 4 patients do not carry any recurrent large-scale chromosomal alterations, but have mutually-exclusive RAS or *FLT3* point mutations, or

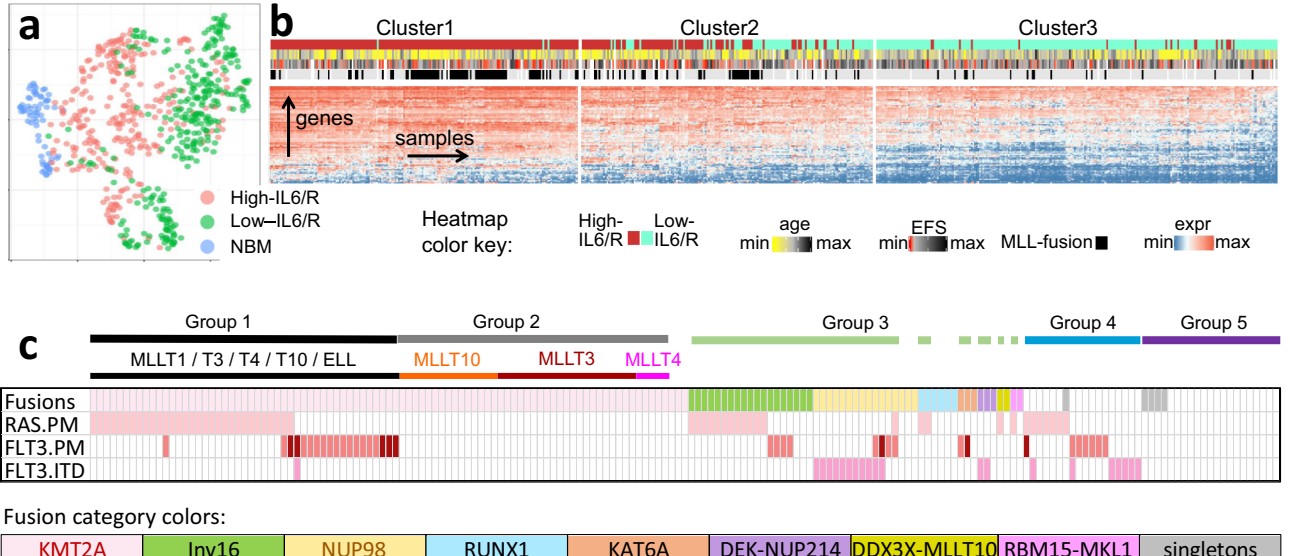

**Fig. 3 | High-IL6/R pAML BM samples are highly distinct in terms of their immunological, transcriptional, and genomic characteristics. a** In a UMAP projection using 1,506 immune genes, 287 high-IL6/R pAML, 306 low-IL6/R pAML and 45 NBM samples form distinct clusters, indicating broad immune differences. **b** Unsupervised hierarchical clustering of the expression levels (log2(TPM + 1)) of 82 immune and cancer genes divides high- and low-IL6/R samples (*n* = 287 and 306 respectively) into 3 groups with sharply distinct rates of *MLL* gene fusions, response to therapy (5-year EFS), and age at diagnosis. The minimum log2 expression value is zero and the maximum is 11.08. The minimum age is 18 days and the maximum is 26.85 years. The minimum EFS is 1 day, and the maximum is 8.2 years. **c** Oncoprint summarizing the distribution of somatic DNA alterations in 181 high-IL6/R pAML diagnostic samples from Cluster1 in (**b**). Each column represents one pAML BM sample. Each row indicates the occurrence of a genomic alteration with a colored cell. Chromosomal translocations are indicated in the "Fusions" row. The fusion partners of *MLL* translocations are indicated above the "Fusions" row. The color key at the bottom shows the colors used to indicate the types of gene-gene fusion displayed in the "Fusions" row (Inv16 = chromosome 16 inversion). The other 3 rows in the Oncoprint indicate the presence of *RAS* and *FLT3* point mutations (PM) and *FLT3* Internal Tandem Duplications (ITD). *FLT3* point mutations that have been experimentally found to activate FLT3 signaling are shown in lighter red. The horizontal bars at the top indicate the grouping of high-IL6/R pAML samples into 5 highly recurrent genomic subgroups. Group1: *MLL*-rearrangements with co-occurring *RAS* or *FLT3* point mutations. Group 2: samples with MLL-rearrangements only. Group 3: samples with diverse gene fusions and co-occurring *RAS* or *FLT3* point mutations. Group 4: samples with *RAS/FLT3* point mutations but no recurrent translocations. Group 5: samples with no detected recurrent genomic alterations.

*FLT3*-ITD. Finally, Group 5 marks 21 high-IL6/R patients (11.6%) with no recurrent somatic alterations in our targeted sequencing data.

Taken together, these findings suggest that high-IL6/R pAML, identified by a distinct transcriptional signature, is driven by specific and highly recurrent genomic alterations, exhibits broadly altered BM immune gene expression, and occurs primarily in younger children and infants.

## Multi-cytokine signaling in high-IL6/R samples is receptor-mediated

We next asked whether the transcriptional signatures enriched in high- versus low-IL6/R pAML are driven by secreted cytokines or arise from intra-cellular gene regulatory interactions driven by genomic alterations. To address these questions, we performed perturbation experiments using high-IL6/R pAML samples.

To mimic in vivo conditions ex vivo, we co-cultured thawed and rested high-IL6/R pAML BM cells in transwells containing the HS-5 stromal cell line in the lower chamber. HS-5 cells replicate BM stromal cell expression patterns, including secretion of IL-1β, IL-6, KIT ligand, and the granulocyte/macrophage colony stimulating factors G-CSF, M-CSF, and GM-CSF[10,20,21]. Cells were cultured with or without the JAK1/2 inhibitor Ruxolitinib (2μM), which has been shown to inhibit signaling by IL-6, IFNs, G/GM-CSF, and TNFα[22]. A 35-marker CyTOF panel (Supplementary Tables 1, 2), designed to measure eight proximal mediators of cytokine signaling in 23 BM cell types (see Supplementary Figs. 18–22), was used to assess proximal signaling 25 min post-stimulation. Additionally, we used bulk RNA-seq to verify downstream pathway activity 7 h post-stimulation.

CyTOF revealed that the leukemic cells of different high-IL6/R pAML genomic subtypes are derived from distinct hematopoietic developmental stages and have distinct signaling states (Fig. 4a, b). As detailed in Supplementary Figs. 23–26, in high-IL6/R samples with MLL gene rearrangements, leukemic cells resembled CD34- pre-monocytes. In contrast, in high-IL6/R samples with chromosome 16 inversions (Inv(16)) or a normal karyotype, the leukemic cells were less differentiated CD34+ common myeloid progenitors (CMP) and granulocyte macrophage progenitors (GMP). In spite of these differences, leukemic cells from all of these samples downregulated the levels of phosphorylated (i.e., activated) STAT1, STAT3, and ERK1/2 following treatment by Ruxolitinib (Fig. 4b, c, Supplementary Data 5). NF-κB, which may be active in unstimulated high-IL6/R samples (Fig. 4b) and is additionally activated by IL-1 and several other mechanisms[23], was not inhibited by Ruxolitinib (Fig. 4c).

Consistent with our CyTOF findings, Gene Set Enrichment Analysis (GSEA) of bulk RNA-seq data from 5 MLL-rearranged high-IL6/R samples and 4 randomly-selected low-IL-6/R samples confirmed downregulation of IL-6, IFNα/β, and TNFα activity by Ruxolitinib in high-IL6/R samples (Fig. 4d). As expected, Ruxolitinib had a lower impact in low-IL6/R samples (Supplementary Fig. 27). Together, our RNA-seq and CyTOF data confirm that high-IL6/R pAML samples can transduce IL-6 (STAT3, ERK1/2), and IFNα/β (STAT1, ERK1/2) signaling, and that active JAK/STAT-mediated signaling is stronger in high- versus low-IL6/R pAML.

## The genomic subtypes of high-IL6/R pAML upregulate distinct pro-leukemia genes

While high-IL6/R pAML samples are genomically diverse (Fig. 3c), ~60% of high-IL6/R pAML subjects have either an MLL-rearrangement with or without *RAS/FLT3* point mutations, or an Inv(16) or *NUP98* translocation with a *RAS/FLT3* point mutation. To determine whether there are common factors driving the high-IL6/R phenotype, we focused on these genomic subtypes. We carried out a series of pairwise differential

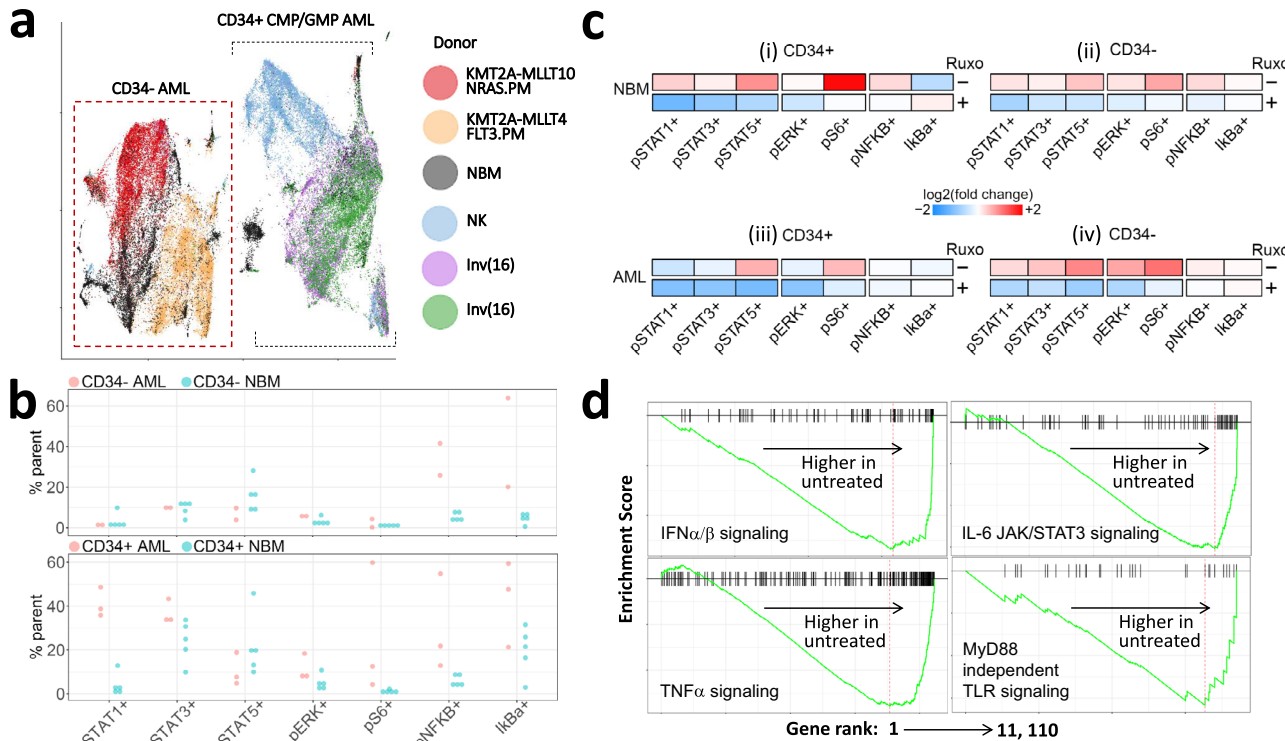

**Fig. 4 | Phospho-CyTOF and RNA-seq suggest distinct cell-of-origin but common receptor-mediated signaling in high-IL6/R pAML genomic subtypes. All plots show data from 5 high-IL6/R pAML samples. Panel A additionally includes 5 NBM as indicated. a** UMAP projection of all CyTOF cell-type markers highlights clustering of cells from 2 samples with *MLL (KMT2A)* fusions away from the cells of 2 samples with chromosome 16 inversion (Inv(16)) and 1 sample (blue markers) with Normal Karyotype (NK). *MLL* fusion partners and *RAS/FLT3* mutation status of the samples are indicated in the color key. The phenotype of each AML cluster is indicated above the cluster. **b** Comparison of the basal signaling states of NBM and the same 5 high-IL6/R samples as in panel (**a**) in the absence of resting in cytokine-rich media or stimulation with HS-5 supernatant. Compared to their counterparts in NBM, CD34+ high-IL6/R pAML samples have higher frequencies of cells with activated (phosphorylated) STAT1 and STAT3, suggesting intrinsic IL-6 and IFNα/β signaling activity. **c** Top row of each panel: Overnight rest in cytokine-rich media, and 25-min stimulation by HS-5 supernatant, increases the frequencies of all signaling-active cells (compared to cells maintained in cytokine-free and HS-5 supernatant-free media) in both NBM (i, ii) and high-IL6/R pAML (iii, iv), except for pSTAT1+ and pSTAT3+ CD34+ CMP/GMP high-IL6/R cells, which are already prevalent in the basal state (see B). Bottom row of each panel: Treatment with Ruxolitinib results in sharp down-regulation of the frequencies of cells with activated JAK/STAT signaling, but has mixed effects on NF-κB signaling. **d** Bulk RNA-seq Gene Set Enrichment Analysis (GSEA) plots of 5 *MLL*-rearranged (i.e., CD34-) high-IL6/R pAML showing downregulation of IFNα/β, TNFα and IL-6 signaling following Ruxo treatment. Consistent with these effects, the MYD88-independent signaling pathway is also downregulated by Ruxolitinib. All GSEA FDR ≤ 0.2.

---

expression analyses (Supplementary Data 6) to identify genes that are (i) differentially expressed in a genomic subset of high-IL6/R pAML compared to low-IL6/R pAML and NBM, (ii) reported to be downstream of the associated genomic alterations, and (iii) reported to regulate the inflammatory signaling pathways co-activated in high-IL6/R pAML. Summary statistics are presented in Supplementary Data 6.

Compared to NBM and low-IL6/R pAML, high-IL6/R pAML samples carrying *RAS/FLT3* point mutations and co-occurring Inv(16) or *NUP98* gene fusions had significantly increased expression levels of the phosphatase *DUSP6*[24–26] (Fig. 5a, b), a known target of *FLT3*-ITD and *RAS* mutations[27,28] and a repressor of DNA damage response[29]. Of note, DUSP6 has been reported to drive resistance to tyrosine kinase inhibitors in pre-B Acute Lymphoblastic Leukemia[30] as well as resistance to cisplatin treatment in solid tumors[31,32], and a recent pre-print[33] reports that elevated DUSP6 acts via the S6 protein to mediate resistance to JAK/STAT inhibition, leading to secondary adult AML. In agreement with this report, we find that untreated pAML samples with upregulated DUSP6 have higher levels of phosphorylated S6 (Supplementary Fig. 26).

In addition to *DUSP6*, high-IL6/R pAML samples with Inv(16) and *RAS/FLT3* mutations had significantly upregulated levels of *miR-221*, a positive regulator of JAK/STAT and NF-κB signaling frequently upregulated in AML[34–36], whereas NUP98-rearranged high-IL6/R pAML had upregulated levels of *STAT5A*, a well-known hematopoietic regulator activated by more than 20 cytokines and growth factors in the BM[37].

In contrast to the above, DUSP6 is not upregulated in MLL-rearranged high-IL6/R pAML. Instead, in MLL-rearranged high-IL6/R samples with and without *RAS/FLT3* mutations, we found *CLEC11A* (also known as stem cell growth factor (SCGF) and osteolectin)[38] and *FLT3* are highly upregulated compared to NBM, and to a lesser extent compared to low-IL6/R pAML samples with matching genomic alterations (Fig. 5c). The role of *FLT3* in pAML is well established[39]. Self-secreted CLEC11A is required for the viability of multiple leukemia cell lines[40], can activate PI3K signaling[41], and has been shown to increase the granulocyte/macrophage colony-forming abilities of G-CSF, GM-CSF, IL-3 and FLT3 ligand[42]. Compared to NBM, *CLEC11A* is expressed at higher levels in low- as well as high-IL6/R samples. However, CLEC11A binds integrin receptors[41,43], and 20 integrin alpha and beta subunit genes, as well as the 'outside-in' integrin signaling licensing genes *TLN1* and *KINDLIN3* are all more highly expressed in high- versus low-IL6/R samples (Supplementary Data 2). Thus, CLEC11A-integrin interactions may have much greater impact in high- compared to low-IL6/R pAML.

A recent analysis of data from The Cancer Genome Atlas (TCGA) suggests *CLEC11A* is epigenetically repressed in adult AML[44,45]. We therefore checked the methylation status of *CLEC11A* in 68 MLL-rearranged high-IL-6/R pAML and 30 NBM samples for which we have assessed genome-wide DNA methylation levels using Illumina Infinium MethylationEPIC arrays. In our pediatric data, the -1kbp *CLEC11A* upstream promoter region is strongly demethylated in *MLL*-

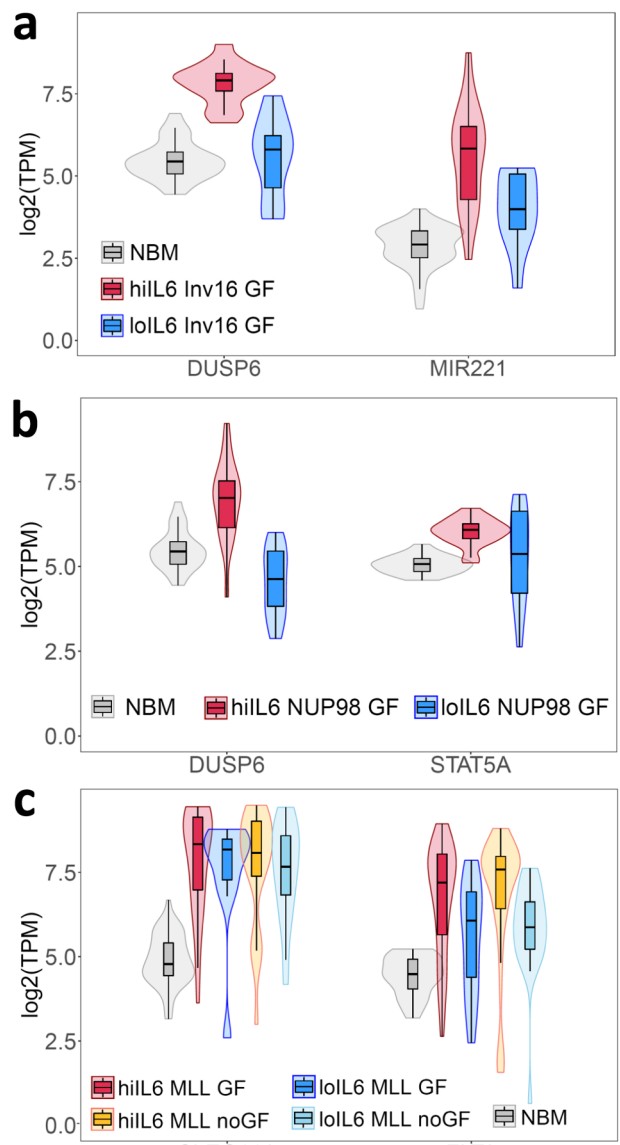

**Fig. 5 | Selected differentially upregulated genes in high-IL6/R pAML samples.** **a** Samples with RAS/FLT3 point mutations and co-occurring Inv(16). **b** Samples with NUP98 translocations and RAS/FTL3 point mutations. **c** Samples carrying ML-rearrangements with or without mutations in the growth factor signaling associated genes FLT3 and RAS ("indicated as "GF" and "noGF" respectively). For all 3 panels, all within-group pairwise two-sided comparison Mann–Whitney $U$ Test $P$ values are <0.05. The abbreviations "hiIL6" and "loIL6" indicate the high-IL-6/R and low-IL6/R pAML subtypes. Numbers of samples per group: NBM = 45, low-IL6/R = 306, hiIL6 Inv16 GF = 17, loIL6 Inv16 GF = 11, hiIL6 NUP98 GF = 20, loIL6 NUP98 GF = 12, hiIL6 MLL GF = 53, loIL6 MLL GF = 13, hiIL6 MLL noGF = 46, loIL6 MLL noGF = 24. Boxplots show the median and the upper and lower central-quartiles. The expected range of the data is indicated by whiskers.

rearranged high-IL6/R pAML compared to NBM, irrespective of the presence or absence of growth factor mutations (Fisher's Exact Test P ~ 1.4E−05, Supplementary Fig. 28). Genome-wide, 7 *CLEC11A*-associated probes were differentially methylated with significantly higher methylation levels in NBM (all FDR-adjusted individual probe moderated *T*-test $P$ values <0.01). Thus, in contrast to adult AML, *CLEC11A* is not epigenetically repressed in MLL-rearranged high-IL6/R pAML and is instead upregulated compared to NBM.

Taken together, the above findings suggest that distinct genes may contribute to poor clinical outcomes in different genomic subtypes of high-IL-6R pAML. Remarkably, all of these differentially

upregulated genes converge on and regulate the inflammatory signaling pathways co-activated in high-IL6/R pAML.

### Higher Toll Like Receptor (TLR) signaling in high versus low-IL6/R pAML

The alarmins S100A8 and S100A9 (hereon S100A8/9) and their receptor TLR4[46,47] are expressed at much higher levels in high-IL6/R pAML samples compared to low-IL6/R pAML samples (Fig. 6). Consistent with this finding, a recent study found that a cluster of 13 of 46 adult AML diagnostic BM samples (28%) had above-median basal expression of TLR4 and 8 and sharply higher expression levels of *S100A8/9, IL-6, IL-1β*, and *TNFα* following TLR4 stimulation with LPS[48].

S100A8/9 expression levels in high-IL-6/R samples do not exceed levels in NBM and are thus not at supra-physiological levels that might be indicative of acute cellular stress. Higher *S100A8/9* expression levels in high- versus low-IL6/R samples are also not due to differences in cellular stresses that might arise from sample processing delays[49] (Supplementary Fig. 29A). Rather, *TLR4* and *S100A8/9* expression appear to be downregulated in low-IL6/R pAML, and co-regulated in high-IL6/R pAML.

In agreement with previous findings in adults[46], in our pediatric samples, *S100A8/9* are expressed in near-perfect correlation across both pAML and NBM samples (Pearson $r = 0.98$, 95% confidence interval: 0.975, 0.982, $P < 2.2E-16$, Supplementary Fig. 29B, C). In addition, *S100A8/9* expression is highly correlated with the expression of *TLR4* (Pearson $r = 0.54$, 95% confidence interval: 0.49, 0.61, $P < 2.2E-16$, Supplementary Fig. 29D, E). These findings suggest common regulatory mechanisms drive *TLR4, S100A8*, and *S100A9* expression in the BM.

Across high- and low-IL6/R pAML samples, TLR4 expression is similarly highly correlated with IL-6 signaling activity (Pearson $r = 0.56$, 95% confidence interval = 0.49–0.63, $P < 2.2E-16$. Supplementary Fig. 30). Moreover, expression levels of the TLRs and their downstream target genes, the Interferon Response Factors (IRFs)[50] sharply segregate high- and low-IL6/R pAML samples (Supplementary Fig. 31), suggesting that *TLR4* expression in high-IL6/R pAML samples is indicative of active TLR signaling in these samples. Taken together, these findings suggest S100A8/9-mediated TLR signaling in high-IL6/R pAML samples accompanies, and may contribute to, IL-6 and IFNα/β signaling activity in these samples.

*S100A8/9* are transcriptionally activated by STAT3 and NF-κB[51-53] and in turn can act as transcriptional co-activators of NF-kB, AP-1 and STAT3[54,55]. In addition, extracellular S100A8/9, released by activated, stressed, and dying neutrophils, monocytes, and macrophages, can bind TLR4[47] and activate IL-6, IL-1, and TNFα signaling[54,56]. Thus, intra- and extracellular S100A8/9 can upregulate each other, and can form a self-reinforcing positive feedback loop that activates and maintains IL-6 signaling.

In adults, high expression of *S100A8/9* has been shown to impact AML severity[57] and treatment resistance[58]. In mice transplanted with AML-inducing cells overexpressing Hoxa9 and Meis1, S100a8 was found to block the differentiation of myeloid-lineage leukemic blasts, and anti-S100a8 antibody significantly improved survival[46].

Activated TLR signaling in high-IL6/R pAML may extend beyond TLR4. Compared to low-IL6/R pAML, high-IL6/R samples have significantly higher expression of 7 TLRs, as well as the TLR-activated interferon response factors (IRFs, Supplementary Fig. 32). In addition to IFNα/β signaling, TLR activation can also directly upregulate IL-6 signaling via increased AP-1 and NF-κB expression[59]. In particular, high expression of *TLR8*, which is the most sharply increased *TLR* in high-IL6/R pAML, has been reported to increase serum IL-6 levels and drive multi-organ inflammation[60]. High-IL6/R samples also have higher expression levels of myeloid and monocytic lineage markers compared to low-IL6/R pAML (Supplementary Figs. 33, 34). Of note, within monocytic and myelomonocytic pAML samples, *IL-6* and *IL-6R*

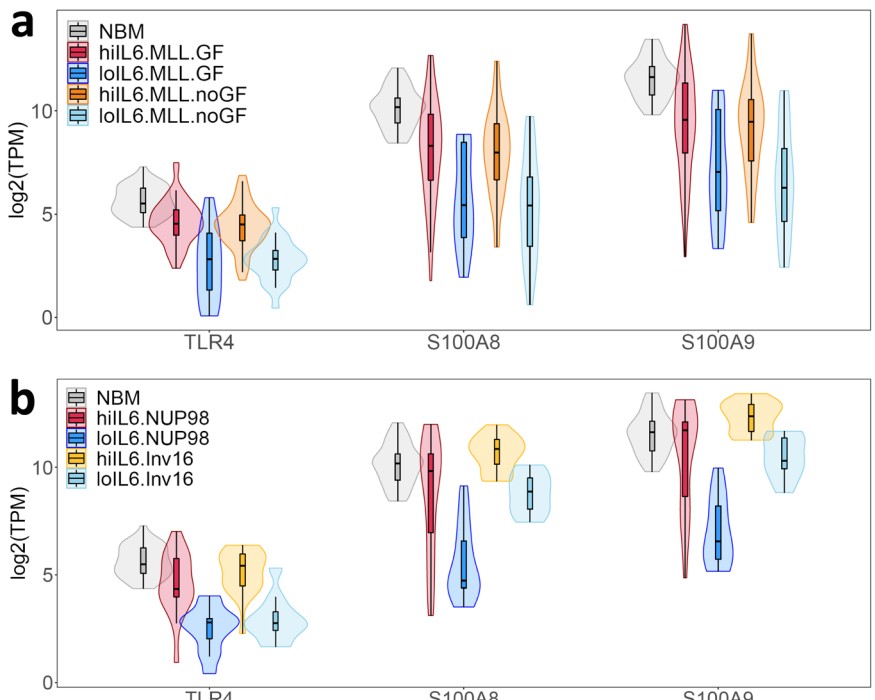

**Fig. 6 | TLR4 and S100A8/9 are expressed at higher levels in high-IL6/R samples compared to low-IL6/R, but not NBM. a** *TLR4* and *S100A8/9* expression levels in MLL-rearranged AML with or without *RAS/FLT3* mutations (indicated as "GF" and"no-GF" respectively), and **b** *TLR4* and *S100A8/9* expression levels in *NUP98*-rearranged and Inv(16) AML with RAS/FLT3 mutations. Numbers of samples per group: hiIL6 Inv16 GF = 17, loIL6 Inv16 GF = 11, hiIL6 NUP98 GF = 20, loIL6 NUP98 GF = 12, hiIL6 MLL GF = 53, loIL6 MLL GF = 13, hiIL6 MLL noGF = 46, loIL6 MLL noGF = 24. Boxplots show the median and the upper and lower central-quartiles. The expected range of the data is indicated by whiskers.

expression levels are much higher in high- versus low-IL6/R samples (Supplementary Fig. 35). Thus, differentiation status or sample composition differences do not explain differences in high- versus low-IL6/R cytokine signaling.

Taking the above observations together, we hypothesize that in high-IL6/R pAML, *S100A8/9* and *TLR* expression may both activate and be activated by IL-6 signaling via NF-κB and AP-1.

## Discussion

Approximately one-third of pAML patients who respond poorly to primary induction therapy die within three years of diagnosis[61]. IL-6 expression in the BM is associated with poor treatment response in both adult and pediatric AML, but its etiology has remained unknown. Using diagnosis stage BM samples from ~1500 pAML patients, we showed here that the IL-6, IFNα,β, IL-1/TLR, and TNFα pro-inflammatory signaling pathways are jointly activated in a subset of pAML patients with poor 2-year outcomes. After adjusting for age and common cytogenetic subtypes, high-IL6/R patients still have worse 2- and 5-year OS (Supplementary Figs. 36–38). Thus, factors beyond age and cytogenetics may contribute to the poor treatment response of this group.

Using a custom CyTOF panel, we showed that the genomic subtypes of high-IL6/R pAML have distinct cells of origin but share the same set of activated receptor-mediated inflammatory signaling pathways. Consistent with our findings, a previous functional screening study found that IL-6, IL-1 and GM-CSF signaling were jointly upregulated in a subset of adult AML patients with diverse genomic and disease subtypes[62].

Remarkably, although each high-IL6/R genomic subtype is associated with high expression of a distinct set of leukemia-driving genes, all these genes activate the same set of signal transduction pathways. Specifically, the 3 gene pairs that are highly expressed in each high-IL6/R pAML subtype—namely *DUSP6* and *STAT5A* in high-IL6/R pAML with

*NUP98* translocations, *DUSP6* and *miR-22*1 in high-IL6/R pAML with Inv(16) translocations, and *CLEC11A* and *FLT3* in high-IL6/R pAML with *MLL*-rearrangements—all activate a common set of pathways: STAT-mediated signal transduction (*STAT5A*, *miR221*, *FLT3*), and PI3K/AKT-mediated signaling (*DUSP6*, *CLEC11A*). Another common feature of all high-IL6/R pAML genomic subtypes is the activation of NF-κB via both PI3K signaling and S100A8/9-mediated TLR signaling.

In addition to context-specific indirect regulation of subtype-specific genes, high-IL6/R associated genomic alterations are also known to directly activate the STAT, PI3K, MAPK, and NF-κB pathways. Most notably, activating *RAS* and *FLT3* point mutations (which co-occur with *MLL*, *NUP98* and Inv(16) translocations in high-IL6/R pAML) are well known to drive the above pathways. Additionally, the MLL protein can upregulate *IL-6* transcription directly via interactions with a long non-coding RNA[63]. It can also bind NF-κB target genes, increasing their expression synergistically[64]. Interestingly, NUP98 fusion proteins physically interact with MLL and transcriptionally regulate many MLL target genes[65], and *NUP98*-fusion xenografts also upregulate IL-6 JAK/STAT3 signaling genes[66].

The fusion protein produced by the Inv(16) translocation has been shown to bind the *DUSP6* gene and upregulate its transcription[67]. In addition, MAPK/PI3K/NF-κB signaling is activated and plays an essential role in Inv(16) AML[68,69].

Signaling via S100A8/9 and TLR4 may be driven by distinct mechanisms in different high-IL6/R subtypes. In *MLL*-fusion AML, signaling via the TLR/IL-1 super-family enhances the chromatin occupancy and downstream effects of MLL-fusion proteins[70]. In addition, the MLL protein (encoded by *KMT2A*) can facilitate NF-κB activity epigenetically[64], leading to higher IL-6 expression[71]. Consistent with these findings, human BM cells engineered to express the MLL-AF9 fusion protein upregulated *TLR4* and *S100A8/9*[72].

Two previous studies provide supporting evidence that in high-IL6/R pAML with *NUP98* and Inv(16) translocations and *RAS/FLT3*

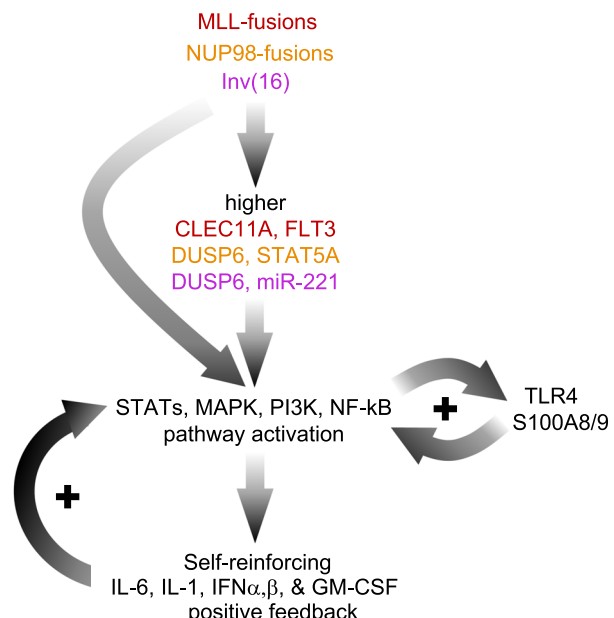

**Fig. 7 | Schematic summary and model of the regulatory interactions reported.** All 3 main genomic subtypes of high-IL6/R pAML upregulate genes that have been reported to activate multiple signal transduction pathways that in turn can upregulate IL-6, IL-1, IFNα/β, GM-CSF, and S100A8/9 mediated TLR signaling. Notably, all these signaling pathways can become self-reinforcing via positive feedback loops (marked by the '+' symbols).

mutations, upregulated *DUSP6* levels may drive TLR signaling and *S100A8/9* transcription. In murine macrophages, DUSP6 was shown to facilitate TLR4/NF-κB driven production of IL-1β and IL-6[73]. In addition, *S100a8* and *S100a9* were both highly upregulated in mouse BM cells engineered to co-express the *Nup98-Nsd1* fusion and an activating RAS mutation[74], the same genomic alterations as in genomic Group 3 high-IL6/R pAML samples, which have upregulated *DUSP6* expression.

In summary, in spite of their genomic and transcriptomic differences, high-IL6/R pAML samples are characterized by a shared set of co-activated signaling pathways (Supplementary Fig. 39). A model of these convergent regulatory interactions is presented schematically in Fig. 7. We note that, once established, such a system of interactions can become self-reinforcing via positive-feedback loops involving secreted S100A8/9, IL-6, IL-1, IFNα/β, and GM-CSF.

Upregulated *IL-6, CLEC11A, DUSP6*, and *S100A8/9* have all been shown to contribute to AML disease severity and treatment resistance. Four additional lines of evidence support the hypothesis that the pathways activated by these genes are key drivers of poor clinical outcomes in high-IL6/R pAML.

First, the IL-6, CLEC11A, DUSP6, and S100A8/9 pathways are the most significantly impacted in terms of high- versus low-IL6/R differentially expressed downstream gene sets. Second, patient stratification based on the expression levels of *IL-6* and *IL-6R* is a better predictor of clinical outcomes than models based on other genes, pathways, and gene sets, suggesting IL-6 signaling is a key contributor to the observed poorer outcomes. Third, the self-reinforcing positive feedback loops created by IL-6, IFNα/β, IL-1 and S100A8/9 signaling in leukemic cells (Fig. 7, Supplementary Fig. 25) can generate highly stable clonal leukemic cell populations that are more likely to become dominant during leukemogenesis. Fourth, the signaling pathways co-activated in high-IL6/R pAML are known to be regulated by the subtype-specific genes highly expressed in high-IL6/R pAML. Thus, multiple redundant pathways may converge-on and drive the activation of cytokine signaling in high-IL6/R pAML. Such "coherent Feed

Forward Loop" network motifs have been shown to be a very common and robust mechanism for responses to perturbations[75,76].

Our study has a number of limitations. All data reported in this study were collected ex vivo from bio-banked samples. Additional regulatory interactions may exist in vivo. Currently, there are no syngeneic mouse models of high-IL6/R pAML, and experiments using fresh (<2 h from extraction) pAML BM samples raise many ethical and operational concerns. In the validation experiments reported here, we approximated the BM microenvironment by using transwells to co-culture thawed biorepository pAML BM samples with stromal HS-5 cells. This approach has two important limitations. First, several cell types present in the BM (e.g., granulocytes, bone tissue) are missing in our experiments. Second, the signaling events of interest occur at different timescales ranging from minutes to hours. Thus, our 25-minute assay provides only a qualitative confirmation of the signaling events and is not a quantitative measure of peak signal activity levels. We also note that the number of samples in our perturbation studies was small. Larger numbers of samples will be needed to confirm the generality of our findings.

We previously reported distinct age-related differences in the patterns of co-occurring and recurrent somatic genomic alterations in pAML[2]. Here we showed that just over 1 in 5 pAML patients who are predominantly infants and children <10 years old have higher activity levels in multiple inflammatory cytokine signaling pathways and poorer treatment responses.

In another recent study[77], we report that infants with AML (defined as being ≤3 years old at diagnosis) also have a distinct BM immune profile. Compared to pAML in children older than 5 years, 95 of the top 200 genes with higher expression in infant-AML are lymphocyte-associated genes (FDR-adjusted Fisher's Exact Test $P \sim 7.7E-13$), and 55 are in the Gene Ontology category "immune system process" (FDR-adjusted Fisher's Exact Test $P$ value $\sim 2.6E-8$). These findings raise the possibility that leukemic immune activity may be a common occurrence in younger children with pAML, and provide a potential clue to the previously observed increased risk of AML in patients with autoimmunity[78].

Taken together, our findings suggest BM leukemic inflammatory signaling is widespread in younger children with pAML, is driven by multiple groups of highly recurrent genomic alterations, and co-activates the JAK/STAT, PI3K, ERK1/2, and NF-kB pathways in all genomic groups. Thus, irrespective of their genomic subtype, high-IL6/R pAML may benefit from therapies that inhibit these co-activated pathways. Effective inhibitors of each of the pathways are already widely available, facilitating the development of combination therapies to inhibit multi-cytokine signaling in high-IL6/R pAML.

## Methods

Pediatric AML biological samples were collected with informed consent (and in accordance with the Declaration of Helsinki) from patients diagnosed with de novo AML and enrolled on Children's Oncology Group (COG) trials AAML0531 (NCT00372593), or AAML1031 (NCT01371981). Each protocol was approved by the National Cancer Institute's central institutional review board (IRB) and the local IRB at Fred Hutchinson Cancer Center (Protocol 9950).

### AAML1031 and AAML0531 RNA-seq

Total RNA was extracted from ficoll-enriched, viably cryopreserved samples from the COG biorepository using the AllPrep Universal Extraction Kit (Qiagen). Nucleic acids were quantified by NanoDrop (Thermo Scientific). RNA samples were tested for quality and integrity using the Agilent 2100 Bioanalyzer (Agilent Technologies). Ribosomal RNA (rRNA) species were removed from total RNA using the NEBNext rRNA Depletion Kit for Human/Mouse/Rat (NEB, E6310X). Strand-specific mRNA libraries were constructed and sequenced at the British

Columbia Cancer Agency's Genome Sciences Centre using an Illumina HiSeq2500 and 75 bp paired-end reads, as previously described[2].

### Ex vivo cell stimulation

Frozen AML cells were thawed in media containing IMDM, 20% BCS, and 100 U/mL DNase I, and rested overnight in media containing 100 ng/mL each of SCF, IL-3, G-CSF, and GM-CSF. Rested cells were transferred to the upper chamber of a transwell with confluent HS-5 cells in its lower chamber with/without added Ruxolitinib. 25 minutes later, an aliquot of the cells was harvested for CyTOF analysis. For RNA-seq, the cells were harvested ~7 h after being transferred to the transwell. HS-5 cells were a gift from Dr. Beverly Torok-Strob (Fred Hutchinson Cancer Center).

### SMART-seq V4 ultra-low-input RNA-seq

Total RNA (0.5 ng) was added to lysis buffer from the SMART-Seq v4 Ultra Low Input RNA Kit for Sequencing (Takara), and reverse transcription was performed followed by PCR amplification to generate full-length amplified cDNA. Sequencing libraries were constructed using the NexteraXT DNA sample preparation kit with unique dual indexes (Illumina) to generate Illumina-compatible barcoded libraries. Libraries were pooled and quantified using a Qubit® Fluorometer (Life Technologies). Sequencing of pooled libraries was carried out on a NextSeq 2000 sequencer (Illumina) with paired-end 59-base reads, using NextSeq P2 sequencing kits (Illumina) with a target depth of 5 million reads per sample. Base calls were processed to FASTQs on BaseSpace (Illumina), and a base call quality-trimming step was applied to remove low-confidence base calls from the ends of reads. The FASTQs were aligned to the GRCh38 human reference genome, using STAR v.2.4.2a (https://github.com/alexdobin/STAR/releases?page=4) and gene counts were generated using htseq-count (https://htseq.readthedocs.io/en/master/install.html). QC and metrics analysis was performed using the Picard family of tools (v1.134).

### Mass cytometry/CyTOF

Details of the antibodies used are given in Supplementary Table 3. To ensure optimal performance, all signaling probes were tested at multiple titrations with BM and blood cells unstimulated and following 25 min of stimulation with IFNB, IL6, LPS, or TCR beads. Upon thaw, cells had high viability (mean 90%, range 78–99%, and retained good viability after overnight resting (mean 79%, range 64–72%). Dead cells were excluded from signaling analyses. There was no association of overall cell viability with the upregulation of signaling.

Dead cells were labeled by incubation in Cell-ID™ Cisplatin (Standard BioTools) solution (5 μM in PBS) for 5 min at 4 °C, then the reaction was quenched and cells washed with addition of an equal volume cold Media (RPMI + 25% FBS). Cells were then resuspended in Media and fixed by addition of and equal volume Fix I Buffer (BD Biosciences) and incubated for 15 min at 37 °C. Cells were next barcoded using Standard BioTools Cell-ID 20-Plex Pd Barcoding Kit, as per manufacturer instructions. After washing, all samples from an individual were combined, and then resuspended in surface staining cocktail in Maxpar® Cell Staining Buffer (CSB, Standard BioTools) for 20 min at room temperature. Combined samples were washed then fixed with 1.6% paraformaldehyde in PBS for 10 minutes at room temperature. Samples were next incubated in ice-cold Perm Buffer III (BD Biosciences) for 1 h at −20 °C, then washed with CSB and then resuspended in intracellular staining cocktail in CSB for 20 min at room temperature. Samples were again washed and fixed with 1.6% paraformaldehyde in PBS for 10 min at room temperature, then stored in Maxpar® Fix and Perm Buffer (Standard BioTools) containing 125 nM Cell-ID™ Intercalator-Ir at 4 °C until acquisition. On the day of acquisition, samples were washed with CSB and then cold ultrapure water, and kept at 4 °C. Immediately before acquisition, samples were resuspended in cold ultrapure water containing 1/5th by volume EQ Four Element Calibration Beads (Standard BioTools) to at target concentration of 1 million cells/mL.

FCS files were normalized and randomized with CyTOF Software (version 7.0.8493) using Uniform Negative Distribution for the randomization and Median Bead Intensity with Passport EQ-P13H2302_ver2 without removing beads for the normalization.

Gating analysis was done using FlowJo version 10.7.1 (BD, flowjo.com). Samples were first debarcoded by gating on cells triple-positive for Palladium isotopes (masses 104, 106, 108, and 110), and individual samples were exported as separate FCS files.

Within each stimulation condition sample, cells were gated to exclude beads (Ce140−), isolate singlets (using Ir191 and Event Length), and capture live cells (Pt198−). Immune cells were identified as live cells that expressed high levels of CD45 but lacked the stromal cell marker CD73. The strategy for identifying stem-cell-like AML blasts was to negatively gate out other immune subsets (Supplementary Data 5). Immune cells were divided into NK cells (CD56+CD3−) and T cells (CD3+CD56−). Within the CD3-CD56- portion, B cells were identified as those expressing CD19 and/or CD20. Non-B cells (CD19-CD20−) were gated to exclude Granulocytes (CD66b+CD15+), and within the non-granulocytes, Monocytes (expressing CD14 and/or CD16) were identified. CD14-CD16- (non-monocytes) were then divided by expression of CD34. Within CD34+ cells, a CD38+ CMP/GMP subset was identified (CMP/GMP AML). Within the CD34− cells, a CD123-CD33+ subset was identified (CD34- AML).

For UMAP projection and probe intensity histograms, signal values were asinh transformed using the Bioconductor (bioconductor.org/) package "CytofKIT" (https://github.com/JinmiaoChenLab/cytofkit) with a co-factor of 5, and analyzed using the Bioconductor packages "flowCore" (https://bioconductor.org/packages/release/bioc/html/flowCore.html), "CytoML" (https://bioconductor.org/packages/release/bioc/html/CytoML.html), and "flowWorkSpace" (https://bioconductor.org/packages/release/bioc/html/flowWorkspace.html). UMAPs were generated using the Bioconductor packages 'SingleCellExperiment' (https://bioconductor.org/packages/release/bioc/html/SingleCellExperiment.html), "scMerge" (https://bioconductor.org/packages/release/bioc/html/scMerge.html) and "scater" (https://bioconductor.org/packages/release/bioc/html/scater.html).

### Statistics

All analyses were performed using R (cran.r-project.org/) and Bioconductor (bioconductor.org/) packages. The analyses reported did not involve any custom statistical procedures, and did not involve any modifications to the libraries/packages used. Unless specifically noted, we used default method/function parameters. *P* values for all multiple-tests were adjusted using the FDR method unless otherwise stated. Bimodal distribution of IL-6 and IL-6R were modeled using the R package "mclust" (https://cran.r-project.org/web/packages/mclust/index.html). Differential expression analyses were performed using the Bioconductor "DESeq2" (https://bioconductor.org/packages/release/bioc/html/DESeq2.html), "edgeR" (https://bioconductor.org/packages/release/bioc/html/edgeR.html), and "limma" (https://bioconductor.org/packages/release/bioc/html/limma.html) R packages. Kaplan-Meier plots and associated *P* values were generated using the "survival" (https://cran.r-project.org/web/packages/survival/index.html) and "survminer" (https://cran.r-project.org/web/packages/survminer/index.html) R packages. Gene Set Enrichment analysis was performed using the Bioconductor package "enrichplot" (https://bioconductor.org/packages/release/bioc/html/enrichplot.html). Unsupervised hierarchical clustering heatmaps were generated using the "pheatmap" (https://cran.r-project.org/web/packages/pheatmap/index.html) and "ComplexHeatmap" (https://www.bioconductor.org/packages/release/bioc/html/ComplexHeatmap.html) R packages. Correlation tests were performed using the R function "corr.test" and the "rcorr" function in the R package "Hmisc" (https://cran.r-project.org/web/packages/Hmisc/index.html).

DNA-methylation analyses were performed using the Bioconductor packages "minfi" (https://bioconductor.org/packages/release/bioc/html/minfi.html), "missMethyl" (https://bioconductor.org/packages/release/bioc/html/missMethyl.html), and "DMRcate" (https://bioconductor.org/packages/release/bioc/html/DMRcate.html) and the "IlluminaHumanMethylationEPICanno.ilm10b2.hg19" (https://bioconductor.org/packages/release/data/annotation/html/IlluminaHumanMethylationEPICanno.ilm10b2.hg19.html), and "IlluminaHumanMethylationEPICmanifest" (https://bioconductor.org/packages/release/data/annotation/html/IlluminaHumanMethylationEPICmanifest.html) annotation packages. Differential methylation results were visualized using the R package "Gviz" (https://bioconductor.org/packages/release/bioc/html/Gviz.html). All other visualizations were generated using the R "ggplot2" (https://cran.r-project.org/web/packages/ggplot2/index.html) package.

### Study approval
All data presented here are derived from COG AAML1031 and COG AAML0531 samples. All patient samples were obtained by member COG institutions after written consent from the parents/guardians of minors upon enrolling in the trial. The study was overseen by the Institutional Review Board at Fred Hutchinson Cancer Research Center (Protocol 9950).

### Inclusion and ethics
All data presented here are from samples collected independently for Children's Oncology Group (COG) studies AAML1031 and AAML0531.

### Reporting summary
Further information on research design is available in the Nature Portfolio Reporting Summary linked to this article.

## Data availability
The raw RNA-seq data for the studies reported here have been deposited in the Database of Genotypes and Phenotypes (dbGaP) under the study ID phs000465.v21.p8. Subject to the NIH Genomic Data Sharing Policy, the raw data are freely available to all researchers via https://www.ncbi.nlm.nih.gov/projects/gap/cgi-bin/study.cgi?study_id=phs000465.v21.p8. Processed data are available at the National Cancer Institute's Genomic Data Commons (https://portal.gdc.cancer.gov/) under the TARGET-AML project. Selected clinical (e.g., age, EFS, OS, cytogenetic classification) and molecular features (e.g., KIT, RAS, NPM1, WT1, CEBPA, IDH1 mutations, and FLT3/ITD allelic ratios) were clinically available for patients included in the NCI/TARGET cohort and are included in the clinical data file available via the TARGET data matrix (ocg.cancer.gov/programs/target/data-matrix). Source data are provided with this paper.

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

## Acknowledgements

The authors are grateful to the children and families who provided samples for COG studies AAML1031 and AAML0531. The generation of multi-omic RNA-seq datasets for these cohorts was funded by the National Cancer Institute (U10CA98543, SM), National Institutes of Health (HHSN261200800001E, SM), Children's Oncology Group Foundation (SM), and the Target Pediatric AML initiative (targetpediatricaml.org, SM). We thank Drs. Anne Hocking, Taylor Lawson, and Virginia Green for their help with manuscript preparation.

## Author contributions

H.B., R.R., and A.E.W. contributed equally to this work. Writing the manuscript: all authors. Data generation and annotation: S.M., R.R., T.H., J.L.S., S.A.L., A.E.W., S.S., V.H.G., K.O., Q.A.N. Study design and analyses: H.B., R.R., A.E.W. Interpretation of analysis results: all authors.

## Competing interests

The authors declare no competing interests.
