## [Peer Review File · Nature Communications]

Inflammatory bone marrow signaling in pediatric Acute Myeloid Leukemia distinguishes patients with poor outcomesREVIEWER COMMENTS

Reviewer #1 (Remarks to the Author): expertise in genomics biostatistics

The manuscript demonstrates that there is co-activation of pro-inflammatory cytokine signaling as a common feature of a large proportion of pAML with diverse genomic subtypes and poor treatment outcomes. A worthy result such as this one that directs our scientific exploration for therapies is a worthy result, no matter if it's "surprising" or not. However, the introduction should give a clearer context enabling the reader to judge the level of surprise I should feel for the finding that co-activation of pro-inflammatory cytokine signaling is a common feature of a large proportion of pAML with diverse genomic subtypes and poor treatment outcomes. The intro covers IL-6, but how surprised should I be to see co-activated cytokine signaling via the IFN, IL-1, and TNF pathways, given what we know about IL-6? In the broadest terms, I don't think it's clearly established by the introduction how important the findings in this manuscript will be. The importance of the general area is well established, but not so much the specifics of the hypotheses / ideas that are addressed in this study.

In the abstract (and in the discussion), it is argued that "Targeted DNA sequencing... revealed ... 5 highly recurrent genomic subtypes". But I think the rigorous process applied to determine these subtypes should be clearer from the beginning (introduction) and throughout (e.g. 1–6). Identifying subtypes, what subdivisions are appropriate to hypothesize, and what is the validation criterion for them? What specific criteria justify the use of "revealed" rather than "selected"? The manuscript should prepare the reader with the decision space within which the manuscript is operating, and ensure that any discretization is supported by rigorous data science.

In the results, it is stated, "IL-6 and IL-6R mRNA levels were bimodally distributed across the pAML cohort": suggesting 2 distinct patient populations (Figure 1B): There is a "dip" between two peaks in these plots, however, such a dip could just be noise. This assertion requires statistical analysis and validation. Of course one can fit a mixture of Gaussians to it, but given the extra parameters required, is that bimodal fit significantly better than appropriate unimodal comparator distribution (gamma, Gaussian)? In both the original and the validation cohort?

Reviewer #2 (Remarks to the Author): expertise in AML genomics

In this study by Bolouri et al entitled "Concurrent IL-6, IL-1, IFN α/β , and TNF α signaling drives poor outcomes in pediatric Acute Myeloid Leukemia", the authors established that the co-activation of pro-inflammatory cytokine signaling in pediatric AML with diverse genetic background. Specifically, the authors show a distinct subset of pAML patients have higher expression of IL-6 and its receptors IL-6R which correlates with poor survival rates. Comparison between pAML samples expressing high and low IL-6/R reveals enrichment of interferon type 1, IL-1, and TNF- α signaling pathways with high IL-6/R expression. The authors also used previously published RNA-seq data from two independent cohorts to strengthen their preliminary findings; however, the statistical significance was not achieved for the survival with the smaller cohort. The authors further characterized the high and low IL-6/R pAML samples with regard to genetic subtypes and found the mutations which have higher representation in high IL-6/R cohort. The authors identify upregulation of many additional signaling pathways in high IL-6/R cohort. The in vitro co-culture of primary bone marrow samples from high IL-6/R pAML patients with HS5 stromal cells and treated with Ruxolitinib showed downregulation of IL-6 and IFN signaling pathways. This study is highly correlative in nature using available RNA-seq data. Establishing these correlations are difficult for such a heterogenous disease. However, it will be great to see supporting validation data showing upregulation of cytokines and pathways and dependency of IL-6/R pAML on these pathways.

Major Concerns:

- Figure 1: The authors selected high IL-6/R group in panel B. It is not clear whether the patients that were selected for this group are expressing both IL-6 and its receptor at high levels or just IL-6 or its receptor. Also showing the correlation between IL-6 and its receptor may be helpful. Figure 1E shows

that high-IL6/R samples are also highly enriched for gene sets marking Type 1 interferon signaling (66 genes), IL-1 signaling (149 genes), and TNF signaling (197 genes). It will be important to show differential expression of all these genes in high and low IL-6/R samples using a heatmap representation to see how expression of each gene changing in each subgroup.

- Figure 2: An independent dataset is used with the same criteria to determine IL-6/R high and low cohorts as in Figure 1. The initial comparison of this cohort to normal bone marrow is missing. Is this dataset also show a bimodal distribution of samples like figure 1?
 - Figure 4: The design of the co-culture experiment with stromal cells expressing different cytokines including IL-6 and IL-1 is not clear. Primary samples were shown to have intrinsic upregulation of IL-6/R. But the co-culture is looking at the extrinsically produced cytokines. The downregulation of inflammatory pathways upon Ruxolitinib treatment is shown, but changes in secreted cytokine levels are not shown.
 - The manuscript describes 60% of high-IL6/R pAML subjects had either an MLL-rearrangement, or an Inv(16) or NUP98 translocation. Then Fig 5 shows the differential expression of selected genes for these genomic subtypes. The difference in CLEC11A expression between high and low IL-6/R seems to be marginal. Compared to the low-IL6/R pAML, high-IL6/R pAML samples carrying RAS/FLT3 point mutations and co-occurring NUP98 gene fusions or Inv(16) had significantly increased level of DUSP6. Do all RAS mutation patients have FLT3 mutations with NUP98 gene fusions or Inv(16) is not clear. In Figure 3C, it seems this is a very small subset with both RAS and FLT3 mutations. Figure 5 does not include comparisons of these sub cohorts with RAS/FLT3 mutations.
 - Throughout the manuscript, a number of correlations have been established. Manuscript could be presented in more focused manner and would be important to show causal relationship for major signaling drivers regulating high IL-6 signaling and how it is regulated by particular genetic subtypes.
- Minor Concerns:
- Figure 3: It is unclear which dataset is used for the analysis in this figure. The definition of minimum and maximum values is missing from panel B.
 - Figure 4E: is the analysis done using DEGs of Ruxolitinib treated vs untreated?
 - Figure 5: typo in panel B. Low IL-6/R is hi-IL6/R
 - Discussion is written with subheading that seems a non-traditional way of putting together discussion for this Journal.
 - It is not clear how many patient samples are involved in each correlation shown throughout multiple figures. P values are missing across multiple correlations.

Reviewer #3 (Remarks to the Author): expertise in inflammatory signalling in AML

Bolouri and colleagues identified a substantial subgroup of pediatric AML patients with high expression of IL-6/IL-6Receptor, that ha a poor clinical outcome. In addition to increased IL-6 levels, also signalling via IL-6, IL-1, TNF α and INF were increased in these patients. This patient group was further characterized by frequent MLL, inv(16), or NUP98 translocations as well as by mutated FLT3 and/or RAS. These abnormalities were associated with increased levels of DUSP6, STAT5, miR221, CLEL11 and/or FLT3, all involved in STAT and PI3K/AKT signalling. Finally, high-IL6/R patients expressed higher levels of TLR4 and S100A8/A9 as compared to low-IL6/R patients. Overall these data may suggest that high-IL6/R patients may profit from therapies targeting multiple cytokine signalling pathways.

This work adds to our understanding of pediatric AML and provides new possible treatment options for such patients. The work is mainly based on RNASeq data, supported by limited cellular analyses. Therefore additional confirmation of the proposed pathways by cellular analyses will be important but this is not necessarily needed for this report.

Comments and suggestions:

AML patients are compared with normal bone marrow samples. How was the processing of these normal samples, Ficoll density centrifugation? Any sorting of myeloid and/or immature populations? Is normal equivalent to healthy, if not please specify normal. For the AML samples which were thawed, was the viability checked before RNA analysis? Some more details in the materials and methods section would help to allow others to reproduce the work.

AML patients are categorized as High IL6/R or low IL6/R. The High group is defined by the highest expressing quartile of IL6 or IL6R. and includes 22.5%. If the selection indeed is based on IL6 or IL6R, this percentage by definition should be >25%. Should "or" be "and"? Similar for the Low group, is the definition based on IL6 or IL6R or IL6 and IL6R?

The heading of the first Results paragraph indicates "Upregulated IL-6 signaling...". Based on the data, IL6-signalling in AML however does not seem to be upregulated as compared to normal bone marrow and therefore "upregulated" may not be the optimal word.

Figure 1B: please add scale for the x-axis to allow easy comparison with Figure 1A. Legend of Figure 1: the last sentence about the abbreviations can be deleted as panel D and E show no abbreviations.

Patients with High IL6/R showed increased IL-6, TNF α , IL-1 and IFN signalling. It might be informative if also some IFN data are shown in Figure 1 (next to extended data in the supplement). Is there any significant relation between the various signalling patterns? Do IL6/R levels related to the signalling levels?

IL6/R expression levels were a better predictor of clinical outcome than IFN or IL1 signalling gene sets. Did the latter two nevertheless have a significant impact on EFS and/or OS?

For the validation cohort, were quartiles used for case classification or the actual levels as obtained in the original cohort? Figure 2 may be moved to the supplement and it may be easier for the reader if the type of signalling is included in the figure itself (panel A,B,C).

Cluster 1 patients had frequent MLL translocations and mutation in RAS and/or FLT3. It may be helpful to also provide these percentages for the two other clusters as well. Same for inv(16) and NUP98.

The 82 genes allowing division between high- and low-IL6/samples includes CD14 and CD4, which might suggest monocytic differentiation of the leukemia or presence of higher numbers of normal monocytes. Did the High-IL6/R patient include more cases with monocytic leukemia?

Signalling was evaluated in co-culture transwell systems, five High-IL6/R patients were used. Differences in phosphorylation seem to occur after Ruxo treatment, but changes are pretty weak, eg pSTA3 (both CD34 AML and GMP AML) and pSTAT1 (CD34 AML). How were such signals in Low-IL6/R patients? Are data shown in panel C from one patient CD34 and one patient CMP/GMP? Figure 4A may read easier if the genetics make-up is shown in the legend in the figure (next to the colored symbols). Are only AML cells shown in the figure?

Only 5 cases were included in the CyTOF analysis.

GSEA of RNASeq data from the co-cultured MLL-rearranged high-IL6/R cases in the presence of Ruxolitinib confirmed downregulation of IL-6, IFN, and TNF α activity. Were similar findings observed in the other 3 cases used in the co-culturing experiment?

The authors focussed on the 60% of high-IL6/R patients with MLL, inv(16) or NUP98 translocation. What was the reason to focus on these and not for example on the larger group of MLL, FLT3 mutated or RAS mutated patients?

Figure 5: panel A and B seem to be swapped (or legend should be adapted).

MLL cases with high-IL6/R are reported to have increased CLEC11A and FLT3 levels, differences in CLEC11 are however very small and there is huge overlap between the various groups.

S100A8/A9 and TLR4 are indicated to be expressed at levels comparable to NBM, but based in Figure 6 levels in AML seems to be significantly lower. The statement “Consistent with this finding, a recent study ..were upregulated...” therefore seems not correct. Rather than upregulated in High-IL6/R cases, it seems TLR/S100A8/A9 is downregulated in Low-IL6/R cases.

High-IL6/R cases have higher expression levels of myeloid and monocytic markers. It would be informative if also CD34, CD117, and CD113 are shown in Supplemental Figure 7A. In Supplemental Figure 7B, Low-IL6/R sample seem to contain many mast cells, which is pretty unusual. How is the mast cell phenotype defined and how do the authors explain this finding?

The authors hypothesize that in high-IL6/R cases, upregulated S100A8/A9 and TLR expression may both activate and be activated by IL-6 signalling. Do the authors already have any data to support this finding? Do IL-6 levels correlate with S100A8/A9 or TLR4?

Reviewer #4 (Remarks to the Author): clinical expertise in paediatric AML

The manuscript by Bolouri, et al asserts that higher expression of IL6 and IL6R genes is a marker of poor outcome in patients with pediatric AML. They find associations of this gene expression pattern with signatures related to other inflammatory genes and signaling pathways, and with specific cytogenetic fusions and mutations. Inflammation in the niche is well-described for AML in elderly adults but has not been extensively studied in pediatric AML, and thus is of interest. Strengths of the manuscript include a very large, well-characterized group of patients from which gene expression data are derived, as well as a relatively large validation sample set from an early clinical trial. A number of weaknesses limit the potential impact of the data, however. For example, there is no validation of the gene expression differences at the protein and functional levels. The only protein / functional data are in the CyTOF experiment, which unfortunately was not designed to demonstrate a difference in signaling between high IL6/R and low IL6/R cases. Another concern is in the validity of the low IL6/R reference group, as the selection criteria were not explained. With uncertainty about the reference group, all subsequent comparisons are questionable. Also, the recurrent observation that the high IL6/R cases are more similar than the low IL6/R cases to the NBM group weakens the argument that high IL6/R expression is aberrant. Additional specific comments and questions follow.

1. Do patients in the “high IL6/R” group have high mRNA levels of both genes, or just one or the other? It would be interesting to see a bivariate plot comparing IL6 v IL6R for the high and low groups.
2. Are there survival differences between high and low IL6 and between high and low IL6R individually? From panel 1A it looks like most of the difference might be driven by high expression of IL6, since this was much less common than high IL6R. Also, what was the reason for using 25% as the cutoff for selection of this group?
3. How were the 306 patients in the “low IL6/R” group selected from the 600 or so with below-median IL6 and/or IL6R mRNA? What steps were taken to ensure that bias was not introduced in the selection of the reference group?
4. IL6ST, which encodes gp130, also is required for the IL-6 signaling complex. How was expression of this gene related to the other two and to survival?
5. Regarding Figure 1C and suppl figure 1, it does not add anything to show 2 yr survival and 5 yr survival separately. The 5 yr curves are nice. Recommend showing the 5 yr curves in Figure 1 and not having a supplemental figure for this point.
6. Figure 1D shows that there are samples in the high IL6/R group with very low blast %. Does that mean that the elevated expression of these two genes was in non-malignant cells in those cases? The same could be said for the low IL6/R group. Since several types of niche cells secrete IL6, how do the authors attribute the differences in gene expression to the AML cells?
7. Along the same lines, Figure 1E and suppl Figure 2 show that the highest enrichment of all 4 inflammatory gene signatures is in the NBM samples. How is that explained?
8. Regarding the caption for Figure 1, the last sentence “In panels D and E, “Hi-IL6” and “Lo-IL6” are abbreviations of “High-IL6/R” and “low-IL6/R” respectively” does not seem to apply.

9. Patient age is discussed at the top of page 7. These authors previously correlated age groups with cytogenetic and molecular findings. Are the same age groups (infant, children, AYA) meaningful in this analysis? For example, is it accurate to say that “infants” <3 years old are enriched in cluster 1?
10. For the mutation analysis, this is a very short list of mutations considered, and several that are clinically relevant and also relevant to inflammation are not included. Recommend broadening the analysis, if data are available, to include at a minimum NPM1, CEBPa, and WT1.
11. To what comparisons do the p values on line 94 refer?
12. The assertion of “broadly disturbed BM immune activity” has not been demonstrated. What was demonstrated was differential expression of immune related genes. Up to this point in the manuscript nothing related to activity has been presented. Recommend clarifying this statement.
13. Regarding the CyTOF perturbation data:
- To conclude that these signaling pathways are more active in high IL6/R cases compared to low IL6/R, it would be important to do these signaling experiments in low IL6/R as well.
 - Since there were NBMs included in the UMAPs, recommend showing the phospho-marker histograms for the normal samples as a comparison.
 - The effect of ruxolitinib on pSTAT3 is rather modest, especially for the CD34- cases. What was the difference in pSTAT3 with and without cytokine perturbation? Was there a positive control to ensure that the HS5 coculture caused the expected changes in signaling? HS5 cytokine secretion can be inconsistent if the cells are not cared for meticulously.
 - Supplemental figure 4B indicates that the UMAPs were constructed from the samples with and without ruxolitinib, and that ruxolitinib had no effect on the distribution of cells over the UMAP. Is that because the UMAPs were constructed from surface markers only, or because the impact of ruxolitinib on the signaling parameters, and thus on overall UMAP distribution, was small?
 - What steps were taken to ensure adequate viability of the samples prior to perturbation? Samples with poor viability post-thaw do not activate signaling pathways.
 - It is curious that the authors show an overlay for pS6 in panel B, since pS6 is not really a focus of these experiments. It would be more relevant to show overlays for pSTAT1 and pSTAT3 in the main manuscript rather than the supplement.
 - Regarding the overlays in suppl figure 4C, it is surprising that pSTAT3 is relatively low in the KMT2Ar cases since these are expected to have high IL6/R expression. Also surprising that ruxolitinib has little perceptible effect on pSTAT5 even though it is activated downstream of ruxolitinib-sensitive JAKs. Can the authors comment?
 - Please clarify how the histograms in supplemental figures 4D and E are different from those in main figure 4.
14. For supplemental table 7, please clarify what the comparison groups are. Does “genomic subset of high IL6/R compared to low IL6R” mean you are comparing, for example, all KMT2A-rearranged cases with all normal karyotype cases? Or KMT2Ar cases with high IL6/r v. KMT2A cases with low IL6/R? The data in Figure 5 suggest it’s the latter but the text is confusing. This is a recurring problem throughout the second half of the manuscript.
15. Figure 5A and B captions do not match the legends.
16. For Figures 5, 6 and all similar gene expression figures, please indicate how many samples are in each group.
17. The difference in CLEC11A expression between high IL6/R and low IL6/R is modest, at least by eye. The fact that it is not differentially expressed in the other high v. low IL6/R comparisons suggests that its overexpression may be more related to the KMT2A fusions, or differentiation state, or any other number of factors.
18. In line 191, TLR4 and S100A8/9 expression is described as “upregulated” but as pointed out in line 184, the expression in the high IL6/R samples is actually similar to NBM and it is the low IL6/R samples that have aberrantly low expression. Please clarify in the text.
19. Please clarify what the comparison groups in supplemental figure 7B are. Are these high and low IL6/R matched for cytogenetics? Otherwise, given the enrichment of KMT2A and inv16 cytogenetics, it is not at all surprising that the high IL6/R samples are enriched for monocytic markers.
20. In the discussion lines 248-161, the text implies that references 64, 67 and 68 were about high IL6/R AML, but in fact they were about AML with cytogenetic findings also seen in the authors’ high IL6/R cases. “High IL6/R” should not be considered synonymous for KMT2A rearranged, for example. Please clarify.

21. Reference #1 is about outcomes after AML relapse, so it is not really an appropriate reference for survival in general. Suggest citing papers reporting the outcomes of cooperative group studies for pediatric AML.
22. The supplemental excel files are not titled in a way that tells the reviewer what they are.

Reviewer #5 (Remarks to the Author): expertise in statistical analysis of -omics datasets

The authors present a series of molecular data analysis results around the concept of signaling pathways for a subset of pediatric AML cases.

Unfortunately the presentation of the data analysis and results are missing common conventions so that the conclusions cannot be fully assessed by the reader. The following needs to be addressed.

1. The statistical analysis section is limited in scope is not sufficient to describe the presented analyses. If there is not room in the manuscript, a supplemental document should be submitted.
2. Any p-value should be listed with it's corresponding test name, e.g., t-test $p=0.006$
3. Any numerical conclusions should be accompanied by estimates with measures of variability of the estimate or a statistical test result, e.g., correlation 0.8, 95%CI 0.65-0.98, increase of expression in group A, t-test $p=0.002$, FDR=0.10. See also text like page 5, line 51 "was a better predictor" which is not accompanied by an estimate for this claim.
4. A consort-like diagram or other method should be presented to demonstrate which subsets of the cases are used in which sections of analysis. This should also show where public sources were added, with their inclusion/exclusion criteria, sample sizes, and access dates.
5. Results of high-throughput screening analyses, from which a single marker is selected for discussion, should be reported first in full. E.g., Of mRNA assessed 1500/25000 were found to be statistically significant (FDR<0.05), among which DUSP was selected for further study due to
6. Conclusions need to be reviewed to ensure that the strength of association aligns with the reported results. E.g., page 5, lines 53-55, it states that an increase in the marker "contributes to poor 2-year outcomes." Yet while the 2-year time point was highlighted in the supplemental figure, no assessment was given for the 2-year survival estimates and no assessment of potential confounders of survival was given. Also the association of a 2-year outcome with treatment stated in the supplement is not demonstrated with the provided data and anyses. More appropriately, it can be said that increased markers are associated with decreased overall survival.

NCOMMS-22-05681, Response to Reviewers

Overall response

We thank the reviewers for their careful reading of the manuscript, and their insightful and constructive suggestions. We have revised our manuscript extensively to fully address all comments. We apologize for the lengthiness of our responses below. It reflects the time and effort the reviewers put in to provide detailed feedback. We feel our manuscript is much improved as a result of us carefully addressing every comment and suggestion.

A Common theme

We apologize for our confusing use of the term “upregulated”. We realize in retrospect that “upregulated” is more naturally interpreted as increased transcriptional activity compared to healthy normal bone marrow (NBM), something we did not intend to imply. In the revised submission, we have edited the text globally to avoid ambiguous use of the term “upregulated”.

Tracking the changes

Line numbers given in the point-by-point responses correspond to locations in the revised manuscript. The first reference to each figure/table is highlighted in cyan. Revised or new sentences cited in our responses below are highlighted in yellow.

Reviewer 1

- 1) *A worthy result such as this one that directs our scientific exploration for therapies is a worthy result, no matter if it's "surprising" or not. However, the introduction should give a clearer context enabling the reader to judge the level of surprise I should feel. I don't think it's clearly established by the introduction how important the findings in this manuscript will be.*

We thank the reviewer for the positive comment and helpful suggestion. We have edited the Introduction (**lines 3-10, new Supplementary Figure 1**) to include a better description of why our findings are noteworthy.

- 2) *In the abstract (and in the discussion), it is argued that "Targeted DNA sequencing... revealed ... 5 highly recurrent genomic subtypes". But I think the rigorous process applied to determine these subtypes should be clearer from the beginning (introduction) and throughout (e.g. 1-6). Identifying subtypes, what subdivisions are appropriate to hypothesize, and what is the validation criterion for them?*

We thank the reviewer for the recommendation, and regret that the manuscript did not fully describe some important details of the targeted genomic sequencing performed, and its implications. We have added explanatory text to the manuscript (**revised Fig. 3C caption, last 5 lines, and lines 108-110, 120-130**).

- 3) *In the results, it is stated, "IL-6 and IL-6R mRNA levels were bimodally distributed across the pAML cohort" [...] Is that bimodal fit significantly better than appropriate unimodal [...]?*

We performed sample selection using both unimodal and bimodal models. Using similar numbers of selected samples for the 2 approaches, we found that the bimodal approach was a better predictor of poor Overall Survival (unimodal $p = 0.0013$, bimodal $p = 0.00049$), as illustrated in **Fig. 1R**. We therefore used the bimodal model in all subsequent analyses.

The bimodal sample selection procedure also had the advantage that it allowed us to simply use the top quartile of the two (IL-6 and IL-6R) higher-expression groups as the samples of interest, whereas the top quartile of the unimodal distribution results in a larger, less distinct population.

We would like to emphasize that we are not proposing that a signature based on the bulk bone marrow expression levels of IL-6 and IL-6R be used as a clinical diagnostic biomarker. Rather, we used IL-6 and IL-6R levels to identify broadly defined groups of patients with and without potential IL-6 signaling in their bone marrows for further analysis. For this reason, we have not included the above analyses in the manuscript.

Fig. 1R. Overall Survival Kaplan Meier plot for high and low IL-6/IL-6R pAML donors selected based on a unimodal sample distribution.

Reviewer 2

- 1) *it will be great to see supporting validation data showing upregulation of cytokines and pathways and dependency of IL-6/R pAML on these pathways.*

There are currently no syngeneic mouse models of high-IL6/R pAML. We hope that our *ex-vivo* findings will motivate future studies using patient-derived xenograft (PDX) models. Within the confines of *ex-vivo* studies, we have re-analyzed our CyTOF perturbation experiments to use cell frequencies instead of signal intensity (new Figures 4 B, C and their captions, and lines 152-161) and show that:

- (a) Unstimulated CD34+ CMP/GMP high-IL6/R pAML cells have higher STAT1 and STAT3 mediated signaling activity compared to their healthy normal bone marrow counterparts (NBM).
- (b) After overnight rest in media containing of SCF, IL-3, G-CSF, and GM-CSF, followed by transwell co-culture with HS-5 cells, CD34- MLL-rearranged high-IL6/R pAML cells also up-regulate JAK/STAT signaling, as do the NBM counterparts of CD34+ and CD34- pAML cells.
- (c) For both CD34+ and CD34- pAML cells as well as their NBM counterparts, the frequencies of cells with activated JAK/STAT1,3,5 signaling decrease sharply following treatment with Ruxolitinib (note log2 scale).
- (d) Ruxolitinib fails to inhibit NF- κ B signaling in high-IL6/R pAML, supporting the hypothesis that in high-IL6/R pAML samples NF- κ B is additionally activated via TLRs, TNF α , and IL-1.
- (e) At the transcriptional level, the extent of JAK/STAT inhibition by Ruxolitinib is greater in high- versus low-IL-/R samples (new Supplementary Figure 18), suggesting stronger JAK/STAT signaling in high-IL6/R samples.

We would like to emphasize that our findings do NOT imply or require that AML blasts initiate and/or maintain IL-6, IFN α/β , or IL-1 mediated signaling in a manner independent of the other cells in the bone marrow environment. Signals via SCF (KIT ligand), Thrombopoietin (TPO), IL-3, IL-6, FLT3 ligand, G-CSF, GM-CSF, and M-CSF are all essential to hematopoiesis and myelopoiesis and are produced and consumed by a variety of bone marrow cells during healthy hematopoiesis. IL-6 in particular is produced in the bone marrow by endothelial cells, osteoblasts, and megakaryocytes, and regulates both HSCs and myeloid progenitors (reviewed in PMC4013788).

The signaling we observe in pAML blasts results from the combined effect of the multitude of cell-cell interactions in the bone marrow. We hypothesize that multiple, overlapping positive feedback loops create a non-linear (and greater than linear) overall effect, but a key feature of such nonlinearity is that it depends on the interactions of all of the components.

2) *showing the correlation between IL-6 and its receptor may be helpful.*

Across all AAML1031 samples, we find that IL-6 and IL6-R transcript levels are not correlated. Only a small proportion of high-IL6/R samples (e.g. 7 of 181 in Cluster1) have high levels of both IL-6 and IL-6R. As a result, following our sample selection procedure (which removes samples with low expression of both IL-6 and IL-6R), IL-6 and IL-6R become negatively correlated.

As shown in Figure 2R, subsets of samples with high (> 70%) and low (< 30%) percent blasts show similar patterns, and the fraction of high-IL6/R samples is the same (~ 20%) in the high and low blast. IL-6 and IL-6R expression levels would be positively correlated if they were both expressed only in a single population (e.g. leukemic blasts). However, IL-6 and IL-6R are expressed by a variety of cells during normal hematopoiesis (PMID:9600782, PMID:14701687). Our results suggest both leukemic cells and also non-leukemic cells (e.g. stroma, lymphocytes) contribute to high levels of IL-6 and IL-6R.

Figure 2R. Relationship between IL-6 and IL-6R mRNA levels (in log₂(TPM) units) across various sample subsets as indicated.

The above observations are consistent with possible IL-6 ‘trans signaling’ in high-IL6/R pAML samples. The protease ADAM17 cleaves membrane-bound IL-6R into a soluble form. The second subunit of the IL-6 receptor, gp130 (encoded by IL-6ST), is widely expressed in stromal and

hematopoietic bone marrow cells [PMID:16118327]. As a result, high IL-6 and ADAM17 expression can result in IL-6 “trans-signaling” in mixtures of cells expressing varying levels of IL-6R. Consistent with this possibility, ADAM17 mRNA levels are high (>16 TPM) in > 96% of high-IL6/R samples (Figure 3R).

Also, consistent with potential IL-6 trans-signaling, the levels of IL-6 and IL-6ST are positively correlated across AAML1031, suggesting replenishment of internalized activated receptor multimers (Pearson $r = 0.31$, 95% confidence interval = 0.264 – 0.360, p -value < 2.2E-16). Of note, the correlation between IL6 and IL6ST expression levels is higher in high-IL6/R pAML samples (Pearson $r = 0.44$, 95% confidence interval = 0.341 – 0.52, p -value = 5.4E-15) compared to low-IL6/R samples (Pearson $r = 0.27$, 95% confidence interval = 0.164 – 0.372, p -value = 1.5E-06), suggesting greater signaling activity in high-IL6/R pAML samples.

IL-6 is produced by a large number of bone marrow cells, including T and B cells, macrophages, megakaryocytes, fibroblasts, endothelial cells, osteoblasts [PMID:14701687]. Thus, availability of soluble IL-6R can result in widespread IL-6 signal transduction in the bone marrow.

Fig. 3R. ADAM17 expression in high-IL6/R AML.

- 3) *Figure 1E shows that high-IL6/R samples are also highly enriched for gene sets marking Type 1 interferon signaling (66 genes), IL-1 signaling (149 genes), and TNF signaling (197 genes). It will be important to show differential expression of all these genes in high and low IL-6/R samples using a heatmap representation to see how expression of each gene changing in each subgroup.*

We thank the Reviewer for the suggestion. We generated a heatmap using multiple gene sets for the Type 1 Interferon, IL-6, IL-1, and TNF α signaling pathways (new Supplementary Figure 6, and associated text on lines 57-58). We show that, IFN α/β , IL-6, IL-1, and TNF α signaling levels are enriched in approximately two thirds of all pAML samples. In addition, the 2 clusters with enrichment for co-expression of IFN1, IL-6, IL-1, and TNF α signaling also have notably higher levels the IL-6 and IL-6 receptor genes (see “IL6 score” column annotation), and are enriched for Cluster1 samples from Figure 3B.

- 4) *Figure 2: The initial comparison of this cohort to normal bone marrow is missing.*

We thank the referee for the suggestion. We have added NBM to Figure 2A-C. In addition, we have added a new Supplementary Figure 12 providing specific examples of the expression levels of selected genes across AAML1031, AAML0531, and NBM.

- 5) *[Does] this dataset also show a bimodal distribution of samples like figure 1?*

Yes. **New Supplementary Figure 11** shows the result of applying exactly the same procedure as for the ‘discovery cohort’ (AAML1031) to the validation cohort (from AAML0531).

6) *Figure 4: The design of the co-culture experiment with stromal cells expressing different cytokines including IL-6 and IL-1 is not clear. Primary samples were shown to have intrinsic upregulation of IL-6/R. But the co-culture is looking at the extrinsically produced cytokines.*

As discussed in our response to the Reviewer 2, Comment 2, our data suggest multiple BM cell types contribute to the increased inflammatory signaling in high-versus low-IL6/R pAML. Healthy bone marrow cells are known to secrete multiple cytokines, including IL-3, GM-CSF, G-CSF, and IL-6. This baseline level of secreted cytokines may be necessary to trigger multi-cytokine signaling by AML blasts and other bone marrow cells. Our co-culture experiments were designed to mimic the supportive *in-vivo* signaling environment *ex-vivo*.

7) *The downregulation of inflammatory pathways upon Ruxolitinib treatment is shown, but changes in secreted cytokine levels are not shown.*

In the *ex-vivo* experiments presented in the manuscript, we show that there is active JAK/STAT signal transduction at the level of post-translational protein modifications downstream of IL-6 and IFN α/β signaling, and that this signaling is inhibited by Ruxolitinib (**revised Figure 4**). We also show in Response to Reviewer 4, Comment 10 (**new Supplementary Figure 18**) that Ruxolitinib inhibits responses to IFN α/β and IL-6 signaling to a greater extent in high- versus low-IL6/R pAML samples.

We did not perform secreted cytokine measurements from bulk Ficoll-enriched frozen samples in part because we felt such measurements would be confounded (i) by being averaged over all cell types, and (ii) by the absence of multiple bone marrow populations that may be producers of the cytokines of interest.

8) *The difference in CLEC11A expression between high and low IL-6/R seems to be marginal.*

We agree with the reviewer that the magnitude of the difference in CLEC11A expression between the high- and low-IL6/R groups is relatively small in log₂ units. However, it corresponds to a 43% difference in TPM units, and is statistically significant (Supplementary Tables 8 I and J). We have edited the relevant sections of the manuscript (**lines 195-197, 203-210, 221-223**) to note that: (i) CLEC11A’s synergistic interactions with IL-3, FLT3 ligand, GM-CSF, and G-CSF may amplify its effects. (ii) the CLEC11A amino acid sequence includes 2 integrin binding motifs and CLEC11A has been shown to bind integrin $\alpha 11$ [PMID: 32003015]. We find that higher expression of CLEC11A in high-IL6/R pAML samples is accompanied by higher expression of 20 integrin alpha and beta subunit genes, as well as the ‘outside-in’ integrin signaling licensing genes TLN1 and KINDLIN3. Thus, CLEC11A-integrin interactions may have much greater impact in high- compared to low-

IL6/R pAML. (iii) CLEC11A may play different roles in adult versus pediatric AML (Supplementary Figure 19).

- 9) *important to show causal relationship for major signaling drivers regulating high IL-6 signaling and how it is regulated by particular genetic subtypes*

We agree with the Reviewer that extensive mechanistic studies will be needed to verify the exact mechanisms by which each of the implicated inflammatory signaling pathways is activated in high-IL-6/R pAML samples. Such studies would require considerable additional effort and are beyond the scope of the current study.

In the current study, we show that: (i) There is considerable literature support for the regulation of these pathways by the somatically altered genes (**summarized in Supplementary Figure 25**). (ii) In the absence of any exogenous stimulation, the 2 different genomic subtypes of high-IL6/R pAML have different basal STAT1/3 signaling levels (**revised Figure 4B**), (iii) The genomic subtypes of high-IL6/R pAML respond differently to cytokine stimulation (**revised Figure 4C**). Thus, different genomic alterations in high-IL6/R pAML samples are associated with distinct JAK/STAT signaling patterns, suggesting a potential causal relationship.

Minor Concerns:

- *Figure 3: It is unclear which dataset is used for the analysis in this figure. The definition of minimum and maximum values is missing from panel B.*

Thank you for the suggestion, and we apologize for the omission. **We have added this information to the Figure 3B caption** (upper highlighted text).

- *Figure 4E: is the analysis done using DEGs of Ruxolitinib treated vs untreated?*

To maximize the number of genes used from each gene set, the analysis was performed using all 11, 110 genes with non-zero expression in the ultra-low input RNA-seq data. **We have added the gene count to the figure.**

- *Figure 5: typo in panel B. Low IL-6/R is hi-IL6/R*

We apologize for the drafting error, now **corrected**.

- *Discussion is written with subheading that seems a non-traditional way of putting together discussion for this Journal.*

Thank you. **We have removed the subheadings.**

- *It is not clear how many patient samples are involved in each correlation shown throughout multiple figures. P values are missing across multiple correlations.*

Thank you. We have added sample numbers to Figures 3, 5 and 6, and added p-values and confidence intervals to all correlations reported in the text, and ensured all comparison populations are stated.

Reviewer 3

- 1) *How was the processing of these normal samples, Ficoll density centrifugation? Any sorting of myeloid and/or immature populations?*

There was no sorting. These are bulk bone marrow samples from healthy donors. Samples were Ficoll-paque enriched for mononuclear cells and frozen.

- 2) *For the AML samples which were thawed, was the viability checked before RNA analysis? Some more details in the materials and methods section would help to allow others to reproduce the work.*

After a rapid thaw, cells are resuspended in PBS and centrifuged to rid the cells of DMSO used in the freezing media. Cells were checked for viability prior to use in the co-culture experiments.

Upon thaw, cells had high viability (mean 90%, range 78-99%, and retained good viability after overnight resting (mean 79%, range 64-72%). Dead cells were excluded. There was no association of overall cell viability with the upregulation of signaling. **We have added this information to the Methods section (lines 405 – 407).**

- 3) *AML patients are categorized as High IL6/R or low IL6/R. The High group is defined by the highest expressing quartile of IL6 or IL6R. and includes 22.5%. If the selection indeed is based on IL6 or IL6R, this percentage by definition should be >25%. Should “or” be “and”? Similar for the Low group, is the definition based on IL6 or IL6R or IL6 and IL6R?*

The number is < 25% because we followed a 2-step selection process. For each of IL-6 and IL-6R, we first divided the samples into high and low expression groups. Next, we used only the groups of samples with higher expression of **either** IL-6 **or** IL-6R, and sub-selected from these samples the top 25% samples with the highest expression of IL-6 **or** IL-6R. Thus, the total number of samples selected is not expected to be 25% of the total.

Of 294 samples selected using the above procedure (23% of all samples), 7 samples were shared between the “high-IL6” and “high-IL6R” groups (i.e. they had high expression of both IL-6 and IL-6R). This left us with a total of 287” high-IL6/R” samples (22.5% of total). **We have edited lines 36-41 of the manuscript to clarify the sample selection procedure.**

- 4) *The heading of the first Results paragraph indicates “Upregulated IL-6 signaling...”. Based on the data, IL6- signalling in AML however does not seem to be upregulated as compared to normal bone marrow and therefore “upregulated” may not be the optimal word.*

We agree, and apologize for our confusing use of the term ‘upregulated’. We have edited the text throughout the manuscript (including headings) to address this issue.

- 5) *Patients with High IL6/R showed increased IL-6, TNF α , IL-1 and IFN signalling. It might be informative if also some IFN data are shown in Figure 1 (next to extended data in the supplement). Is there any significant relation between the various signalling patterns? Do IL6/R levels related to the signalling levels?*

Thank you. We have re-organized Figure 1 to include an IFN1 enrichment plot in panel (E). To highlight the extent of correlated activity among these signaling pathways, we generated a heatmap using multiple gene sets for the Type 1 Interferon, IL-6, IL-1, and TNF α signaling pathways (new Supplementary Figure 6, and associated text on lines 57-58). We show that, IFN α/β , IL-6, IL-1, and TNF α signaling levels are enriched in approximately two thirds of all pAML samples. In addition, the 2 clusters with enrichment for co-expression of IFN1, IL-6, IL-1, and TNF α signaling also have notably higher levels the IL-6 and IL-6 receptor genes (see “IL6 score” column annotation), and are enriched for Cluster1 samples from Figure 3B.

- 6) *IL6/R expression levels were a better predictor of clinical outcome than IFN or IL1 signalling gene sets. Did the latter two nevertheless have a significant impact on EFS and/or OS?*

Independently, IFN1 and IL-1 are each worse predictors of EFS and OS than selection using IL6 and IL6R, as detailed in the new Supplementary Figures 7 to 10 and added lines 59-64.

It should be noted that instead of using IFN1/IL-1 ligand and receptor expression levels to select samples, we used multiple published IL-1 and IFN1 signaling gene sets, and performed single-sample Gene Set Enrichment Analysis (ssGSEA) because (i) IL1 α/β are post-transcriptionally modified into activating or inhibitory ligands via proteolysis. As a result, transcriptional levels of IL1 α/β may not be robust predictors of IL-1 signaling. (ii) The IL-1 receptor dimer IL1R1:IL1RAP can be inhibited by expression of IL1R2, and the extent of this inhibition is affected by common IL1R1 and IL1R2 polymorphisms (iii) There are 16 different Type 1 Interferon genes (including IFNB1 and 12 IFNA genes) with different receptor binding affinities [PMC7850986]. (iv) The receptors IFNAR1 and IFNAR2 are ubiquitously expressed, making their bulk mRNA levels uninformative.

- 7) *Did the High-IL6/R patient include more cases with monocytic leukemia?*

They do. Thank you for the suggestion. We have added this information on lines 275-284 of the revised text.

8) *Signaling was evaluated in co-culture transwell systems, five High-IL6/R patients were used. Differences in phosphorylation seem to occur after Ruxo treatment, but changes are pretty weak, eg pSTA3 (both CD34 AML and GMP AML) and pSTAT1 (CD34 AML).*

We realize in retrospect that our CyTOF total signal intensity histogram plots in the earlier version of this figure were overly complicated and difficult to interpret. Accordingly, **we have modified Figure 4 of the manuscript, its caption, and lines 148-157** to provide a fuller and more intuitive presentation using signaling-active cell frequencies. Please also see our response to Reviewer 2, Comment 1.

9) *How were such signals in Low-IL6/R patients? Are data shown in [Fig 4] panel C from one patient CD34 and one patient CMP/GMP?*

Due to sample and resource limitations we did not run CyTOF on low-IL6/R samples. At the transcriptional level, we show in the **new Supplementary Figure 18** that Ruxolitinib also reduced these signals in low-IL6/R samples, but to a lesser extent than in high-IL-6R samples, suggesting stronger signaling in the latter group.

Regarding the high-IL6/R patient subtypes represented in Figure 4B-C, the samples are the same as in panel A, and were split into 2 groups: 3 samples with CD34+ CMP/GMP blasts, and 2 samples with MLL-rearrangements and CD34- monocytic-lineage blasts (as indicated in Figure 4A). For RNA-seq, we needed >3 samples in order to gain sufficient statistical power. Thus, the RNA-seq GSEA plots in Figure 4D are for 5 MLL-rearranged high-IL6/R samples combined.

10) *GSEA of RNASeq data from the co-cultured MLL-rearranged high-IL6/R cases in the presence of Ruxolitinib confirmed downregulation of IL-6, IFN, and TNFa activity. Were similar findings observed in the other 3 cases used in the co-culturing experiment?*

Due to sample volume and availability issues, the RNA-seq experiments did not use the same samples as the CyTOF experiments. For the GSEA analysis presented in Figure 4D, to ensure sufficient statistical power and minimize inter-sample variability, we focused on MLL-rearranged samples (which comprise ~50% of high-IL6/R samples).

11) *The authors focussed on the 60% of high-IL6/R patients with MLL,, inv(16) or NUP98 translocation. What was the reason to focus on these and not for example on the larger group of MLL, FLT3 mutated or RAS mutated patients?*

The selected samples include samples carrying MLL-rearrangements with and without activating FLT3 and RAS point mutations (Groups 1 and 2 in Figure 3C, respectively), as well as samples with other common pAML translocations and co-occurring RAS/FLT3 point mutations (members of Group 3 in Figure 3C). We focused on these 60% of high-IL6/R samples because for each genomic subtype in these samples, we have a sufficiently large number of samples to allow statistically robust subtype comparisons.

12) MLL cases with high-IL6/R are reported to have increased CLEC11A and FLT3 levels, differences in CLEC11 are however very small and there is huge overlap between the various groups.

We agree with the reviewer that the magnitude of the difference in CLEC11A expression between the high- and low-IL6/R groups is relatively small in log₂ units. However, it corresponds to a 43% difference in TPM units, and is statistically significant (Supplementary Tables 8 I and J). We have edited the relevant sections of the manuscript (**lines 195-197, 203-210, 221-223**) to note that: (i) CLEC11A's synergistic interactions with IL-3, FLT3 ligand, GM-CSF, and G-CSF may amplify its effects. (ii) the CLEC11A amino acid sequence includes 2 integrin binding motifs and CLEC11A has been shown to bind integrin α 11 [PMID: 32003015]. We find that higher expression of CLEC11A in high-IL6/R pAML samples is accompanied by higher expression of 20 integrin alpha and beta subunit genes, as well as the 'outside-in' integrin signaling licensing genes TLN1 and KINDLIN3. Thus, CLEC11A-integrin interactions may have much greater impact in high- compared to low-IL6/R pAML. (iii) CLEC11A may play different roles in adult versus pediatric AML (**Supplementary Figure 19**).

13) S100A8/A9 and TLR4 are indicated to be expressed at levels comparable to NBM, but based in Figure 6 levels in AML seems to be significantly lower.

We apologize for miscommunicating. The statement:

"Of note, all 3 genes were expressed at levels comparable to NBM, implying physiological levels of activity."

was intended to point out that S100A8/9 and TLR4 levels are NOT being expressed at supra-physiological levels that might be indicative of acute cellular stress. **We have changed the wording (lines 231-235) to clarify the statement.**

14) The statement "Consistent with this finding, a recent study ...were upregulated..." therefore seems not correct.

This is a continuation of the above miscommunication. The cited research reports that a subset of adult AML samples with a monocytic (FAB M4/M5) classification and higher than median expression levels of TLRs 4 and 8 responded to LPS stimulation with sharply higher expression

levels of S100A8/9, IL-6, IL-1 β , and TNF α . Consistent with this report, high-IL6/R pAML samples have a monocytic profile and express higher levels of S100A8/9 and TLR4. S100A8/9 compared to low-IL6/R pAML samples. We show that S100A8/9 expression levels are highly correlated with that of TLR4 (Supplementary Figure 20), suggesting that they are predominantly expressed in the same cells (presumably monocytes). Taken together, these findings suggest monocyte-like cells in high-IL6/R samples may be producing S100A8/9 and responding to these ligands via TLR4. **We have amended our text (lines 225-235) to clarify the above issues.**

15) Rather than upregulated in High-IL6/R cases, it seems TLR/S100A8/A9 is downregulated in Low-IL6/R cases.

We apologize for the inappropriate use of the term “upregulated”. **We have edited our text (lines 236-249)** to clarify our reasoning that there is active S100A8/9 mediated TLR signaling in high-IL6/R samples.

16) *High-IL6/R cases have higher expression levels of myeloid and monocytic markers. It would be informative if also CD34, CD117, and CD113 are shown in Supplemental Figure 7A.*

CD34, KIT/CD117, and PROM1/CD133 are each positive in subsets of both high- and lowIL6/R pAML samples. The expression pattern of these genes in the bone marrow is complicated because they are expressed at multiple points during normal hematopoiesis (PMIDs 10723580, 9711908, and 32242051 respectively). We find that their inclusion does not change the distinct high- and low-IL6/R sample clustering by myeloid markers, as shown in **Figure 4R** (cf. Supplemental Figure 23A). We did not include these genes in Supplementary Figure 23A to avoid confusing readers.

Figure 4R. CD34, KIT (CD117) , and PROM1 (CD133) are expressed on subsets of high- and low-IL6/R pAML samples and do not impact clustering by myeloid lineage markers.

17) In Supplemental Figure 7B, Low-IL6/R sample seem to contain many mast cells, which is pretty unusual. How is the mast cell phenotype defined and how do the authors explain this finding?

CYBERSORTx predicts cell type labels based on gene expression signatures derived from healthy blood samples. Therefore, the cell type labels generated for our pAML bone marrow samples should be viewed as suggesting expression profiles most resembling the indicated blood cell types. Thus, cells labeled “Mast” may be considered more broadly as granulocyte-like cells, probably Granulocyte/Macrophage lineage cells. We included this figure to show that the proportions of the major cell populations may be dramatically different between high- and low-IL6/R pAML.

To reflect the above considerations and avoid mis-interpretation of the plots by readers, **we have simplified the cell type labels in Supplementary Figure 23C. We have also added explanatory text to the figure caption.**

18) The authors hypothesize that in high-IL6/R cases, upregulated S100A8/A9 and TLR expression may both activate and be activated by IL-6 signalling. Do the authors already have any data to support this finding? Do IL-6 levels correlate with S100A8/A9 or TLR4?

Thank you for the suggestion. Yes, our data suggest IL-6 signaling activity is correlated with TLR4 expression. As there are no individual unambiguous markers of IL-6 signaling activity, we estimate IL-6 signaling activity using the median expression of the Gene Ontology IL-6 signaling gene set. As shown in our new Supplementary Figure 21, TLR4 expression and IL-6 signaling are strongly correlated (Pearson $r = 0.56$, 95% confidence interval = 0.49-0.63, p -value < 2.2E-16). Supporting the idea of co-signaling by IL-6, TLR4 and IFN α/β , we see strong correlation between the expression levels of TLRs and IRFs, and a near perfect segregation of high- and low-IL6/R samples by these genes (new Supplementary Figure 22). We have added a paragraph (lines 243-249) to report these findings.

Other Comments:

- *For the validation cohort, were quartiles used for case classification or the actual levels as obtained in the original cohort?*

The validation cohort samples were selected independently of discovery cohort. We applied the exact same sample selection procedure to both the discovery (AAML1031) cohort and the validation cohort.

- *it may be easier for the reader if the type of signalling is included in the figure itself (panel A,B,C).*

Thank you for the suggestion. Done.

- *Cluster 1 patients had frequent MLL translocations and mutation in RAS and/or FLT3. It may be helpful to also provide these percentages for the two other clusters as well.*

Thank you for the suggestion. We have added the MLL-rearrangement frequency in Cluster3 samples (7%) to the text (line 100). RAS and FLT3 mutation rates in Cluster3 are: RAS point mutations 17%, FLT3 point mutations 8%, FLT3 ITD 18%. We have added RAS and FLT3 Cluster1 vs. Cluster 3 mutation enrichment statistics to the manuscript lines 112-113.

- *Only 5 cases were included in the CyTOF analysis*

This study analyzes one of the largest pAML RNA-seq datasets to date ($n = 1489$), and includes DNA-methylation analysis for 98 samples, and targeted DNA sequencing of 181 samples. The focused *ex-vivo* experiments reported here provide independent validation of our findings in a small number of randomly selected samples. We hope that sharing the mutually-supportive findings from these complementary large- and small-scale analyses will enable their further validation by the AML research community and fuel progress towards in-human clinical trials.

Reviewer 4

- *(Preamble Comment 1) There is no validation of the gene expression differences at the protein and functional levels. The only protein / functional data are in the CyTOF experiment, which unfortunately was not designed to demonstrate a difference in signaling between high IL6/R and low IL6/R cases.*

There are currently no syngeneic mouse models of high-IL6/R pAML. We hope that our *ex-vivo* findings will motivate future studies using patient-derived xenograft (PDX) models. In an *ex-vivo* approximation of the bone marrow environment, we show at the protein level that there is active multi-cytokine inflammatory signaling in high-IL6/R samples and that this activity can be inhibited by targeted drugs (**revised Figure 4**). In additional material presented in response to your comment 10 (**new Supplementary Figure 18**), we also show results suggesting the magnitude of response to JAK/STAT inhibition is higher in high-IL-6/R pAML samples compared to low-IL6/R pAML samples, suggesting stronger JAK/STAT signaling in high-IL6/R samples.

- *(Preamble Comment 2) Another concern is in the validity of the low IL6/R reference group, as the selection criteria were not explained.*

We apologize for the lack of clarity, and agree with the review that unbiased selection of the reference group is important. To avoid selection biases, we defined the low-IL6/R reference group simply as the intersection of samples with below-median expression of (**both**) IL-6 and IL-6R across all pAML samples. We would like to point out that our conclusions do not change if we instead use the low-IL6/R cluster of samples identified in Fig. 3B (Cluster3) as our reference 'low-IL6/R' group, as illustrated in **new Supplementary Figures 14A-C, cited on line 97**.

- *(Preamble Comment 3) Also, the recurrent observation that the high IL6/R cases are more similar than the low IL6/R cases to the NBM group weakens the argument that high IL6/R expression is aberrant.*

We realize in retrospect that our phrasing was confusing and thank the reviewer for pointing this out. **We have edited the text globally to avoid giving the impression that IL-6 signaling is aberrant.**

In addition to inflammatory signaling, the transcriptomic profiles of high- and low-IL6/R pAML samples are dramatically different. High-IL6/R pAML transcriptomes are also very different from NBM transcriptomes (**lines 85-91, Supplementary Tables 1 and 2**). Many inflammatory signaling genes play important roles in normal healthy hematopoiesis by regulating processes such as proliferation, differentiation, and egress from the bone marrow. Thus, the context in which inflammatory signaling takes place in high-IL6/R samples is very different from NBM. We hypothesize that – **in the context of the altered transcriptomes in high-IL6/R pAML** – inflammatory signaling (a known driver of treatment resistance) contributes to poorer outcomes.

1) Do patients in the “high IL6/R” group have high mRNA levels of both genes, or just one or the other? It would be interesting to see a bivariate plot comparing IL6 v IL6R for the high and low groups.

Our sample selection procedure for the high-IL6/R group combines samples with high expression of either IL-6 or IL-6R. For the low-IL6/R group, we selected samples with the expression levels of both IL-6 and IL-6R below their median across the whole AAML1031 cohort. Figure 5R below summarizes high- versus low-IL6/R expression differences in IL-6 and IL-6R as requested.

Figure 5R. Comparisons of the expression levels of IL6 and IL6R in the high-IL6/R and low-IL6/R pAML patient groups.

2) Are there survival differences between high and low IL6 and between high and low IL6R individually? From panel 1A it looks like most of the difference might be driven by high expression of IL6, since this was much less common than high IL6R. Also, what was the reason for using 25% as the cutoff for selection of this group?

We thank the Reviewer for the astute observation. To address this issue, we selected equal numbers of highest and lowest expressing samples for each of IL-6 and IL6R. Using a conservative selection threshold (n = 144 per group), the 5-year log-rank p-values for Overall and Event-Free Survival (OS and EFS) are:

For the high versus low IL-6 expression comparison, OS p = 0.16, EFS p = 0.079
 For the high versus low IL-6R expression comparison, OS p = 0.014, EFS p = 0.0044

Choosing n = 288 (similar to the size of the high-IL6/R group) only changed the p-values slightly:

For the high versus low IL-6 expression comparison, OS p = 0.16, EFS p = 0.16
For the high versus low IL-6R expression comparison, OS p = 0.007, EFS p = 0.0018

Thus, selection using both IL-6 and IL-6R improves predictions, especially of OS. Regarding our choice of selecting the top 25% from each of the high-expression IL-6 and IL-6R subsets of patients, we intentionally did not search for an “optimal” threshold to avoid over-fitting the data. We simply selected the top-quartile of each population.

3) How were the 306 patients in the “low IL6/R” group selected from the 600 or so with below-median IL6 and/or IL6R mRNA? What steps were taken to ensure that bias was not introduced in the selection of the reference group?

As noted above, we tried to avoid bias by defining the reference group very broadly and simply as samples with below-median expression of **both** IL-6 and IL-6R. We then tested the robustness of our findings by using a different reference group selected on the basis of unsupervised hierarchical clustering (Cluster3 of Fig 3B). As illustrated in our **new Supplementary Figure 14**, there is more than 90% agreement among genes differentially expressed between healthy normal bone marrow and either the low-IL6/R samples used in the manuscript, or the low-IL6 samples in Cluster3 of Fig. 3B.

4) IL6ST, which encodes gp130, also is required for the IL-6 signaling complex. How was expression of this gene related to the other two and to survival?

Across all AAML1031 samples, IL6ST expression is slightly negatively correlated with IL-6R (Pearson $r = -0.11$), but positively correlated with IL-6 expression (Pearson $r = 0.30$). These correlation levels are essentially unchanged in low-IL6/R samples (-0.04 and 0.27 respectively). The correlations become sharper in high-IL6/R samples. For IL-6 and IL6ST, $r = -0.33$, and for IL6 and IL6ST, $r = 0.44$.

Regarding survival analysis using IL-6ST, selecting 288 samples with the highest and lowest expression levels of IL-6ST, the log rank p-value for EFS = 0.16, and for OS = 0.012.

5) Regarding Figure 1C and suppl figure 1, it does not add anything to show 2 yr survival and 5 yr survival separately. The 5 yr curves are nice. Recommend showing the 5 yr curves in Figure 1 and not having a supplemental figure for this point.

Thank you for the suggestion. **We agree that the p-values in Figure 1C should be for the 2-year period and have updated the figure accordingly (ditto Figure 2 p-values, line 73).** We feel showing both time spans is important because poor 2-year EFS and OS is suggestive of resistance to therapy and because the largest differences in outcomes between high- and low-IL6/R pAML patients arise at this timepoint. With the current arrangement, we provide full 5-year details in the Supplementary Figure while focusing the reader on the key message in Figures 1 and 2.

6) *Figure 1D shows that there are samples in the high IL6/R group with very low blast %. Does that mean that the elevated expression of these two genes was in non-malignant cells in those cases?*

Of 287 high-IL6/R (Figure 1D), 8 samples are missing percent blast values. Of the remaining 279, 23 samples (8.2%) contain less than 20% blasts. In these cases, the high levels of IL-6/IL-6R could be due to non-leukemic cells, a mixture of leukemic and non-leukemic cells, or very high expression levels in the relatively small blast population. Unfortunately, the bulk nature of our RNA-seq, does not allow us to distinguish among these scenarios.

6B) *Since several types of niche cells secrete IL6, how do the authors attribute the differences in gene expression to the AML cells?*

We agree with the reviewer that multiple populations in the bone marrow express IL-6. The observed IL-6 signal transduction is most likely happening in leukemic blasts, because these cells are the predominant population in our samples (see Figure 1D and preceding response). Supporting this hypothesis, we show that transcriptional targets of IL-6 signal transduction (IL-6 gene sets from GO and MSigDB) are expressed at higher levels in high-IL6/R samples compared to low-IL6.R samples. The IL-6 gene is a transcriptional target of IL-6 signaling in many cell types, an autocrine effect sometimes referred to as the IL-6 amplifier (PMID: 33337480). We therefore believe it is likely that leukemic bone marrow cells in high-IL-6/R samples are engaged in autocrine IL-6 signaling, probably via ADAM17-mediated ‘trans’ signaling (see response to Comment 2 by Reviewer 2).

7) *Along the same lines, Figure 1E and suppl Figure 2 show that the highest enrichment of all 4 inflammatory gene signatures is in the NBM samples. How is that explained?*

In addition to inflammatory signaling, the transcriptomic profiles of high- and low-IL6/R pAML samples are dramatically different. High-IL6/R pAML transcriptomes are also very different from NBM transcriptomes (revised manuscript lines 85-91, Supplementary Tables 1 and 2). Many inflammatory signaling genes play important roles in normal healthy hematopoiesis by regulating processes such as proliferation, differentiation, and egress from the bone marrow. Thus, the context in which inflammatory signaling takes place in high-IL6/R samples is very different from NBM. We hypothesize that – in the context of the altered transcriptomes on high-IL6/R pAML – inflammatory signaling (a known driver of treatment resistance) contributes to poorer outcomes.

8) *Regarding the caption for Figure 1, the last sentence “In panels D and E, “Hi-IL6” and “Lo-IL6” are abbreviations of “High-IL6/R” and “low-IL6/R” respectively” does not seem to apply.*

We apologize for the drafting error, now corrected.

9) *Patient age is discussed at the top of page 7. These authors previously correlated age groups with cytogenetic and molecular findings. Are the same age groups (infant,*

children, AYA) meaningful in this analysis? For example, is it accurate to say that “infants” <3 years old are enriched in cluster 1?

We thank the reviewer for the suggestion. We have added a figure (**Supplementary Figure 15**) and included a note in the text (**lines 103-105**) to highlight the enrichment in the infant age group.

The age annotation color scale in the heatmap the reviewer refers to is already configured such that ‘yellow’ corresponds approximately to the infant age group (≤ 3 years). In **Figure 6R** below, we have added an additional annotation row with discretized age groups for infant (≤ 3 years), children (4-15 years), and adolescents and young adults (15-39 years). Given the high similarity between the discretized and continuous color scales, we have retained the more high-resolution continuous color scale in the manuscript.

10) For the mutation analysis, this is a very short list of mutations considered, and several that are clinically relevant and also relevant to inflammation are not included. Recommend broadening the analysis, if data are available, to include at a minimum NPM1, CEBP α , and WT1.

We have **added lines 108-110** to explain that we assessed targeted sequencing data for a panel of 580 recurrent single nucleotide alterations and short insertion/deletions (indels), and 15 translocations previously identified by the NCI TARGET AML Project. Only significantly recurrent alteration patterns are reported in the manuscript.

NPM1, CEBP α , and WT1 were included in our targeted sequencing panel. However, their mutation frequencies in the high-IL6/R group are not higher than expected in general. WT1 occurs in 4.4% of high-IL6/R samples, but in 8.3% of pediatric AML [PMID: 20413658]. NPM1 mutations happen in only 1.7% of high-IL6/R samples, but they occur in 7.6% of pediatric AML patients [PMID:31915364]. CEBP α also occurs in 4.4% of high-IL6/R samples. Its frequency in pediatric AML in general is 4.5% [PMID: 19304957]. Interestingly, the frequency of CEBP α mutations is higher in the low-IL6/R group (see column annotation in **Figure 6R**).

11) To what comparisons do the p values on line 94 refer?

We apologize for the omission. The comparison group was the Cluster3 low-IL6/R samples from Figure 3B. **We have added this information to the text (now on line 113).**

Figure 6R. The heatmap of Figure 3B reproduced with additional column annotations including pre-defined age groups, and example genomic aberrations.

12) *The assertion of “broadly disturbed BM immune activity” has not been demonstrated. What was demonstrated was differential expression of immune related genes. Up to this point in the manuscript nothing related to activity has been presented. Recommend clarifying this statement.*

We agree with the Reviewer’s distinction between transcriptional activity and protein activity. Approximately one third of all immune genes used in Figure 3A are differentially expressed between NBM and high-IL6/R pAML samples, suggesting global immune differences. Accordingly, **we have modified the text (lines 132-133)** to read “exhibits broadly altered BM immune gene expression”.

13) *To conclude that these signaling pathways are more active in high IL6/R cases compared to low IL6/R, it would be important to do these signaling experiments in low IL6/R as well.*

We thank the reviewer for the suggestion. We quantified low-IL6/R sample responses to Ruxolitinib in our initial RNA-seq studies. We have now **added 2 figures (Supplementary Figure 18, lines 160-161)** to show that high-IL6/R samples respond more strongly to JAK/STAT inhibition compared to low-IL-6R samples.

14) *[Reviewer’s Comment 13b] Since there were NBMs included in the UMAPs, recommend showing the phospho-marker histograms for the normal samples as a comparison.*

We have included NBM cell frequencies in the revised Figure 4.

15) *[Reviewer’s Comment 13c] The effect of ruxolitinib on pSTAT3 is rather modest, especially for the CD34- cases. What was the difference in pSTAT3 with and without cytokine perturbation? Was there a positive control to ensure that the HS5 coculture caused the expected changes in signaling? HS5 cytokine secretion can be inconsistent if the cells are not cared for meticulously.*

We thank the Reviewer for the insightful observations and helpful suggestions, which led us to realize that the signal intensity histogram plots in Figure 4 were overly complicated and difficult to interpret. We have switched our analyses to use cell frequencies instead of total signal intensities. The frequencies of signaling-active cells with/without Ruxolitinib treatment provide a much clearer picture of the effect of Ruxolitinib on all 3 STATs. **We have revised Figure 4 of the manuscript accordingly.** For additional comments, please see our response to Reviewer 2, Comment 1.

16) *[Reviewer’s Comment 13d] Supplemental figure 4B indicates that the UMAPs were constructed from the samples with and without ruxolitinib, and that ruxolitinib had no effect on the distribution of cells over the UMAP. Is that because the UMAPs were constructed from surface markers only, or because the impact of ruxolitinib on the signaling parameters, and thus on overall UMAP distribution, was small?*

We thank the Reviewer for spotting this drafting error. The caption of Figure 4A should have said that the UMAPs were generated using all Cytof **cell type** markers (not “all 35 CyTOF markers”). We apologize for the error and have corrected the caption of Figure 4A accordingly.

17) [Reviewer’s Comment 13e] *What steps were taken to ensure adequate viability of the samples prior to perturbation? Samples with poor viability post-thaw do not activate signaling pathways.*

Upon thaw, cells had high viability (mean 90%, range 78-99%, and retained good viability after overnight resting (mean 79%, range 64-72%). Dead cells were excluded. There was no association of overall cell viability with the upregulation of signaling. We have added this information to the Methods section (lines 405 – 407).

18) [Reviewer’s Comment 13f] *It is curious that the authors show an overlay for pS6 in panel B, since pS6 is not really a focus of these experiments.*

Thank you for the suggestion. We have removed the pS6 overlay plot from the manuscript and instead integrated it into Supplemental Figure 17D.

19) [Reviewer’s Comment 13g] *Regarding the overlays in suppl figure 4C, it is surprising that pSTAT3 is relatively low in the KMT2Ar cases since these are expected to have high IL6/R expression. Also surprising that ruxolitinib has little perceptible effect on pSTAT5 even though it is activated downstream of ruxolitinib-sensitive JAKs. Can the authors comment?*

We apologize for the lack of clarity and have **revised Figure 4 B, C** of the manuscript to address the issues raised here. Please see our response to your point 15 above. In particular, we note that in terms of cell frequencies, there is a clear STAT1/3/5 response to Ruxolitinib in all samples.

20) [Reviewer’s Comment 13h] *Please clarify how the histograms in supplemental figures 4D and E are different from those in main figure 4.*

Figure 4 has now been revised to show cell frequencies. Supplemental Figure 17 now shows all the intensity histograms for completeness.

21) [Reviewer’s Comment 14] *For supplemental table 7, please clarify what the comparison groups are. Does “genomic subset of high IL6/R compared to low IL6R” mean you are comparing, for example, all KMT2A-rearranged cases with all normal karyotype cases? Or KMT2Ar cases with high IL6/r v. KMT2A cases with low IL6/R? The data in Figure 5 suggest it’s the latter but the text is confusing. This is a recurring problem throughout the second half of the manuscript.*

Thank you for the suggestion. We compare each high-IL6/R subset first to ALL low-IL/R samples and then to all NBM samples. We have added explanatory captions to each of the sub-tables A-M in Supplementary Table 8 (was 7).

22) [Reviewer's Comment 15] *Figure 5A and B captions do not match the legends.*

Thank you for the correction. The legends to Figure 5 A and B were swapped. This has now been corrected.

23) [Reviewer's Comment 16] *For Figures 5, 6 and all similar gene expression figures, please indicate how many samples are in each group.*

Thank you for the suggestion. These numbers have been added to the captions of Figures 3–6.

24) [Reviewer's Comment 17] *The difference in CLEC11A expression between high IL6/R and low IL6/R is modest, at least by eye.*

We agree with the reviewer that the magnitude of the difference in CLEC11A expression between the high- and low-IL6/R groups is relatively small in log2 units. However, it corresponds to a 43% difference in TPM units, and is statistically significant (Supplementary Tables 8 I and J). We have edited the relevant sections of the manuscript (lines 195-197, 203-210, 221-223) to note that: (i) CLEC11A's synergistic interactions with IL-3, FLT3 ligand, GM-CSF, and G-CSF may amplify its effects. (ii) the CLEC11A amino acid sequence includes 2 integrin binding motifs and CLEC11A has been shown to bind integrin $\alpha 11$ [PMID: 32003015]. We find that higher expression of CLEC11A in high-IL6/R pAML samples is accompanied by higher expression of 20 integrin alpha and beta subunit genes, as well as the 'outside-in' integrin signaling licensing genes TLN1 and KINDLIN3. Thus, CLEC11A-integrin interactions may have much greater impact in high- compared to low-IL6/R pAML. (iii) We present evidence (Supplementary Figure 19) CLEC11A may play different roles in adult versus pediatric AML.

25) [Reviewer's Comment 18] *In line 191, TLR4 and S100A8/9 expression is described as "upregulated" but as pointed out in line 184, the expression in the high IL6/R samples is actually similar to NBM and it is the low IL6/R samples that have aberrantly low expression. Please clarify in the text.*

We agree with the Reviewer and regret the miscommunication. We have re-written lines 225-249 and made minor changes to lines 250-266 to correct this miscommunication.

26) [Reviewer's Comment 19] *Please clarify what the comparison groups in supplemental figure 7B are. Are these high and low IL6/R matched for cytogenetics? Otherwise, given the enrichment of KMT2A and inv16 cytogenetics, it is not at all surprising that the high IL6/R samples are enriched for monocytic markers.*

The samples randomly selected because selecting high- and low-IL6/R samples with matching cytogenetics would lead to a severely biased comparison due to the extensive genomic differences between the 2 groups.

We agree with the reviewer's point that high-IL6/R samples may be expected to be monocytic. **We have updated lines 275-284** of the manuscript to clarify this point. The Supplementary Figure is one of 2 provided in support of this expectation.

27) [Reviewer's Comment 20] In the discussion lines 248-161, the text implies that references 64, 67 and 68 were about high IL6/R AML, but in fact they were about AML with cytogenetic findings also seen in the authors' high IL6/R cases. "High-IL6/R" should not be considered synonymous for KMT2A rearranged, for example. Please clarify.

We apologize for poorly communicating this point. We did not intend to imply that high-IL6/R and MLL-rearranged pAML are synonymous. Our intention was to report literature support for our findings. **We have corrected the sentence citing this reference** (formerly 64, now 70), **and clarified the related text (lines 309-326).**

28) [Reviewer's Comment 21] Reference #1 is about outcomes after AML relapse, so it is not really an appropriate reference for survival in general. Suggest citing papers reporting the outcomes of cooperative group studies for pediatric AML.

We thank the referee for the correction. **We have replaced the citation** with a reference to a study spanning all pediatric AML patients and reporting 10-year outcomes. **The text has been corrected accordingly (line 3).**

29) [Reviewer's Comment 22] The supplemental excel files are not titled in a way that tells the reviewer what they are.

Thank you for pointing this out. **We have added explanatory captions to the Supplementary tables.**

Reviewer 5

- 1) *The statistical analysis section is limited in scope is not sufficient to describe the presented analyses. If there is not room in the manuscript, a supplemental document should be submitted.*

Thank you for the suggestion. We have expanded the "Statistics" sub-section of Methods (lines 415-419, 421-427) to include a comprehensive list of all R and Bioconductor packages used. We have also explained that (a) the analyses reported did not involve any custom statistical procedures, and did not involve any modifications to the libraries/packages used, and (b) unless specifically noted, we used default method/function parameters.

- 2) *Any p-value should be listed with it's corresponding test name, e.g., t-test $p=0.006$*

Thank you for the suggestion. Done.

- 3) *Any numerical conclusions should be accompanied by estimates with measures of variability of the estimate or a statistical test result, e.g., correlation 0.8, 95%CI 0.65-0.98, increase of expression in group A, t-test $p=0.002$, FDR=0.10. See also text like page 5, line 51 "was a better predictor" which is not accompanied by an estimate for this claim.*

Thank you for the suggestion. We have added correlation confidence intervals and new Supplementary Figures 7-10 to support the assertion that selection using IL6/IL6R provided better predictions of survival.

- 4) *A consort-like diagram or other method should be presented to demonstrate which subsets of the cases are used in which sections of analysis. This should also show where public sources were added, with their inclusion/exclusion criteria, sample sizes, and access dates.*

We thank the reviewer for the suggestion. We have added a CONSORT-like diagram (Supplementary Figure 2). No inclusion/exclusion criteria were applied beyond those of the parent clinical trials.

Although the samples we analyze in this manuscript were collected for Children's Oncology Group (COG) clinical trials, the analyses presented in this manuscript are not part of any clinical trial. Rather, FHCRC-IRB approved protocol 9950 allowed us to obtain previously collected specimens from COG for further study and characterization. As such, our study did not involve recruitment, randomization, and other common features of CONSORT flow diagrams.

- 5) *Results of high-throughput screening analyses, from which a single marker is selected for discussion, should be reported first in full. E.g., Of mRNA assessed 1500/25000 were*

found to be statistically significant (FDR<0.05), among which DUSP was selected for further study due to

Thank you for the suggestion. Because we used a 3-step process to identify subtype-specific genes, there are 5 numbers to report per subtype (total 20). **To aid flow and readability, we have collated these data into a new Supplementary Table 8N.** We describe our methodology for selecting specific genes for discussion on lines 168-172.

6) *Conclusions need to be reviewed to ensure that the strength of association aligns with the reported results. E.g., page 5, lines 53-55, it states that an increase in the marker "contributes to poor 2-year outcomes." Yet while the 2-year time point was highlighted in the supplemental figure, no assessment was given for the 2-year survival estimates and no assessment of potential confounders of survival was given.*

Thank you for the suggestions. **We have updated Figure 1C and the text associated with Figure 2D (lines 73-74) to show the 2-year p-values.** Readers will likely want to see 5-year curves and p-values as well. We have therefore included the full 5-year data in Supplementary Figure 3.

Regarding confounders of 2-year survival, we have **added lines 45-48** to note that donor sex and rates of participation in AAML1031 Trial Arms were similar between high- and low-IL-6/R samples. Other factors are discussed at various points in the manuscript, including: blast percentage (**line 45**), age at diagnosis (**102-105**), genomic abnormalities (**106-130**), and FAB classification (**275-284**).

6B) Also the association of a 2-year outcome with treatment stated in the supplement is not demonstrated with the provided data and analyses. More appropriately, it can be said that increased markers are associated with decreased overall survival.

We have **globally modified our text as well as the manuscript title** to avoid giving the impression that we have demonstrated a causal link. We have also edited the text globally to clarify that poorer 2-year survival *may* be indicative of treatment resistance.

While we present considerable indirect evidence that inflammatory signaling in the bone marrow drives poor short-term event-free and overall survival in pAML, we agree with the reviewer that only in-human perturbation experiments (usually clinical trials) can provide evidence of causality.

REVIEWER COMMENTS

Reviewer #1 (Remarks to the Author):

The revisions have addressed the concerns I expressed in my original review.

Reviewer #2 (Remarks to the Author):

Authors addressed most of my previous comments, manuscript is much more improved in the analysis and rationals.

To strengthen the significance of the study, it would have been great to see experimental validation as mentioned in my prior comment showing the dependency on signaling pathway such as IL6r signaling for highIL-6R/pAML. May be this could be done by performing in vitro culture using primary samples with IL6r CRISPR or blocking antibodies, and quantifying the effects on HSC and myeloid cells by flow cytometry and colony formation assays, in the absence of transgenic models.

Reviewer #3 (Remarks to the Author):

Thank you for your revisions, my comments have sufficiently been addressed.

Reviewer #4 (Remarks to the Author):

The authors have clearly made substantial and thoughtful revisions based on the comments of the reviewers. My remaining questions and concerns have to do mostly with Figure 4, which is new, and Figure 5/Supplemental Table 8, which I still do not quite understand. Overall, I will again note that almost all the figures and tables were generated from existing RNA-seq data sets. The only data toward validation of a potential inflammatory phenotype at the protein and functional levels are in Figure 4. It is unfortunate that no low IL6/R samples were studied by CyTOF as this significantly limits the only functional experiment. There are lots of associations but no demonstration that relatively higher expression of IL6/R is causally related to poor outcomes. Therefore this manuscript consists primarily of hypothesis generation rather than hypothesis testing. The last sentence of the results section confirms as much. Specific comments follow:

- Page 4 line 32 “collected at diagnosis prior to diagnosis” should be “...prior to treatment”??
- Page 4 line 38 – The authors noted in the response letter that only 7 of 287 patients had high expression of both IL6 and IL6R. This is noteworthy and deserves to be stated in the initial description of the population. Explaining this will also really help the reader understand many of the figures.
- Text related to Figure 1E – Please clarify that the “enrichment” in these signatures for the high IL6/R group is relative to the low IL6R group. Compared to the NBM samples there is actually relative depletion.
- Figure 4C – Please indicate what these frequency fold change values are relative to. Are the values for Ruxolitinib-treated cells relative to the same condition, or relative to the stimulated condition?
- Supplemental Table 8 DEG comparisons – These are still somewhat baffling to me. If I understand the description in Table 8N, the tables are comparing high IL6/R cases with (for example) KMT2Ar against all low IL6/R cases regardless of cytogenetic background. But the figures in Figure 5 suggest the comparisons are for high IL6R v low IL6/R with matched genomic features. It is important to know exactly what is being compared before the results can be interpreted. Ultimately, it is not at all clear what these comparisons add to the more straightforward comparison already presented in suppl table 2. In the absence of validation, especially at the protein and functional levels, one should be cautious about drawing conclusions. For example, in the rebuttal the authors state that the log₂FC in CLEC11A mRNA expression amounts to a 43% difference. But does this mean anything at all in terms of protein expression or function?
- Discussion p. 16 line 282 – There is a sentence of new text stating that within the subset of samples

with M4/5 morphology, IL6 and IL6R mRNA levels are higher in the high IL6/R cases than the low IL6/R. This point is rather self-evident. I wonder if the point the authors want to make is that not all M4/5 cases have high IL6/R expression, so the mRNA profile is not simply explained by differentiation status. Would also suggest making this point in the Results section, rather than presenting more data in the Discussion. A similar point about differentiation status is also made with the cytof data.

Reviewer #5 (Remarks to the Author):

To the authors: Thank you for adding the additional supplemental tables and annotating reported tests and estimates.

Thank you for including the R packages used to conduct your analysis plan. Please be sure to add the appropriate citations for those packages into your reference list. While a tool list does not explain the analysis plan, I found that analysis details were added to figure legends and the text.

Thank you for adding the graphical summary of the datasets used. The reference to a CONSORT diagram was only to illustrate a type of flow diagram that show's inclusion-exclusion. The phrase "consort-like" can be dropped from the descriptions of this figure.

I agree that the full survival curves should be presented in figures 1, 2, and S3. The request was not to truncate the curves to 2-years of follow-up but to provide the point estimate of proportion surviving at 2 years in each group, with 95%CI of those estimates. From figure S3a, this looks like about 35% vs 55%. Please re-instate the full survival curves, while noting specific estimates with 95%CI at times of interest. Please add y-axis labels on figure 2D.

Regarding adjusted survival, age differs by group and age at diagnosis has been shown to be a significantly associated with mortality. Molecular classifications, as shown in Fig 3 and lines 275-284, are not given for the IL6-low group. Molecular classes are also associated with survival. Please conduct an adjusted model to determine if age and molecular classification affects the survival outcomes and the influence of the IL6-high classification.

Point-By-Point Response to Reviewer Comments

Reviewer #4 (Remarks to the Author):

Thank you for making the suggested clarifications and for answering my questions. I have no further comments.

Response: We thank the reviewer.

Reviewer #5 (Remarks to the Author):

Thank you for conducting the adjusted survival analyses with age and genomic group. While age is not strongly associated with these survival outcomes, it appears that some of the genomic groups are

Response: Thank you. We agree.

The only critique is a small adjustment to the new wording in lines 73-78. Since a log-rank test is not for any single point in time but equally weights all points under consideration, please make the following edit: Consistent with a role of for IL-6 in treatment resistance, (delete: two-year) EFS and OS were significantly worse for high-IL6/R patients compared to low-IL6/R patients log-rank $P = 0.028$ and 0.014 respectively, (add: with) proportion of high-IL6/R alive at 2 years = 0.512 , 95% Confidence Interval = $(0.392, 0.700)$, proportion of low-IL6/R alive at 2 years = 0.738 , 95% Confidence Interval = $(0.633, 0.861)$).

Response: We have made the changes specified by the Reviewer (underlined above). The text now reads:

“Consistent with a role of for IL-6 in treatment resistance, ~~two-year~~ EFS and OS were significantly worse for high-IL6/R patients compared to low-IL6/R patients log-rank $P = 0.028$ and 0.014 respectively, with proportion of high-IL6/R alive at 2 years = 0.512 , 95% Confidence Interval = $(0.392, 0.700)$, proportion of low-IL6/R alive at 2 years = 0.738 , 95% Confidence Interval = $(0.633, 0.861)$.”